# Re-Evaluating the Oxidative Phenotype: Can Endurance Exercise Save the Western World?

**DOI:** 10.3390/antiox10040609

**Published:** 2021-04-15

**Authors:** Filip Kolodziej, Ken D. O’Halloran

**Affiliations:** Department of Physiology, School of Medicine, College of Medicine & Health, University College Cork, T12 XF62 Cork, Ireland; K.OHalloran@ucc.ie

**Keywords:** oxidative stress, endurance exercise, metabolic disease, oxidative phenotype

## Abstract

Mitochondria are popularly called the “powerhouses” of the cell. They promote energy metabolism through the tricarboxylic acid (TCA) cycle and oxidative phosphorylation, which in contrast to cytosolic glycolysis are oxygen-dependent and significantly more substrate efficient. That is, mitochondrial metabolism provides substantially more cellular energy currency (ATP) per macronutrient metabolised. Enhancement of mitochondrial density and metabolism are associated with endurance training, which allows for the attainment of high relative VO_2_ max values. However, the sedentary lifestyle and diet currently predominant in the Western world lead to mitochondrial dysfunction. Underdeveloped mitochondrial metabolism leads to nutrient-induced reducing pressure caused by energy surplus, as reduced nicotinamide adenine dinucleotide (NADH)-mediated high electron flow at rest leads to “electron leak” and a chronic generation of superoxide radicals (O_2_^−^). Chronic overload of these reactive oxygen species (ROS) damages cell components such as DNA, cell membranes, and proteins. Counterintuitively, transiently generated ROS during exercise contributes to adaptive reduction-oxidation (REDOX) signalling through the process of cellular hormesis or “oxidative eustress” defined by Helmut Sies. However, the unaccustomed, chronic oxidative stress is central to the leading causes of mortality in the 21st century—metabolic syndrome and the associated cardiovascular comorbidities. The endurance exercise training that improves mitochondrial capacity and the protective antioxidant cellular system emerges as a universal intervention for mitochondrial dysfunction and resultant comorbidities. Furthermore, exercise might also be a solution to prevent ageing-related degenerative diseases, which are caused by impaired mitochondrial recycling. This review aims to break down the metabolic components of exercise and how they translate to athletic versus metabolically diseased phenotypes. We outline a reciprocal relationship between oxidative metabolism and inflammation, as well as hypoxia. We highlight the importance of oxidative stress for metabolic and antioxidant adaptation. We discuss the relevance of lactate as an indicator of critical exercise intensity, and inferring from its relationship with hypoxia, we suggest the most appropriate mode of exercise for the case of a lost oxidative identity in metabolically inflexible patients. Finally, we propose a reciprocal signalling model that establishes a healthy balance between the glycolytic/proliferative and oxidative/prolonged-ageing phenotypes. This model is malleable to adaptation with oxidative stress in exercise but is also susceptible to maladaptation associated with chronic oxidative stress in disease. Furthermore, mutations of components involved in the transcriptional regulatory mechanisms of mitochondrial metabolism may lead to the development of a cancerous phenotype, which progressively presents as one of the main causes of death, alongside the metabolic syndrome.

## 1. Introduction

Nearly one hundred years ago, Hill, Long, and Lupton [1] published their seminal work on “*Muscular exercise, lactic acid, and the supply and utilisation of oxygen*”. They observed increases in ventilation (V_E_), oxygen consumption (VO_2_), and blood lactate concentration ([La^−^]) in response to incremental exercise, which spurred the concept of aerobic capacity (VO_2_ max) and gave rise to the field of exercise physiology. Since the validation of VO_2_ max as an ultimate measure of oxidative/endurance performance [2,3], several critical or “threshold” phenomena have been described. These include ventilatory threshold (VT), lactate threshold (LT) [4], anaerobic threshold (AnT) [5], the onset of blood lactate accumulation (OBLA) [6], and max lactate steady state (MLSS) [7]. More popular in recent literature is the focus on peak fat oxidation (PFO) and coinciding work rate (Fat Max) [8,9]. The inter-individual variations in these critical phenomena are manifested in the level of endurance performance such as in middle-distance running, marathon, triathlon, road cycling, cross-country skiing, and many more. These threshold values occur at separate intensities within one individual, and they are subject to improvement with consistent endurance training. The first purpose of this review is to elucidate the metabolic pathways and mechanisms that translate to oxidative performance in skeletal muscles. Secondly, the signalling mechanisms associated with endurance exercise are outlined, and their role in the development of a superior oxidative phenotype is discussed. We highlight the role of mitochondria, which are the oxidative entities of cells that drive the adaptive signals in response to the stressful stimulus of exercise. We hypothesise that the foundations of endurance exercise adaptation are deeply based on the principles of the endosymbiotic theory of eukaryotic evolution, where the oxidative symbiont responds to disruptions in homeostasis (caused by exercise) by rendering the system more oxidative, therefore capable of meeting the demands of the similar exercise stimulus upon next exposure. Consequently, these are manifested in improved critical metabolic measurements (Fat Max and LT).

## 2. Rate of ATP Hydrolysis at Myofilaments Creates Demand

The exercise performance, whether measured in terms of speed, power output, or VO_2_, is only a tip of the iceberg, which includes the biomechanics of movement [10] and the consequent work economy (ability to maintain given work rate with relatively smaller VO_2_) [11]. However, more importantly from the standpoint of this review, the crucial component comes from the capacity of working muscles to replenish the ATP pool during continuous contractions. As the mechanical work rate is elevated with neural stimulation, oxygen consumption rises in a logarithmic fashion until it plateaus at a new steady state. This phenomenon has been coined an “oxygen deficit” [12]. Three energy systems can replenish the ATP shortage (from fastest to slowest to react): phosphagen, glycolytic phosphorylation, and oxidative phosphorylation (OXPHOS). In the phosphagen system, ATP is recycled by donation of a phosphoryl group from phosphocreatine (PCr) to ADP by a creatine kinase (CK)-catalysed reaction [13]. The PCr pool is limited and can regenerate ATP for only several seconds of high work rate. Continuous regeneration of PCr is essential to muscle contractile function and homeostatic recovery. PCr levels tend to overshoot during recovery from a high-power output exercise bout, a trait associated with high cardiorespiratory fitness [14]. The second most responsive system is glycolysis, which generates two moles of ATP per molecule of glucose without engaging mitochondrial metabolism. This anaerobic pathway can sustain ATP recycling for up to several minutes as marked by short-lived efforts above the anaerobic threshold [15]. Last to respond is the sluggish OXPHOS system. For years there has been debate as to whether the O_2_-deficit is the result of delayed oxygen delivery to contracting muscles or the lag in the activity of mitochondrial enzymes of OXPHOS. Recent advances suggest that the latter is the dominant factor [16]. Therefore, in prolonged endurance exercise, the most important factor for the sustainability of performance is the capacity to swiftly respond to any increments in demand and steadily oxidise the carbon substrates (carbohydrates, fatty acids, amino acids, and lactate) for long-term ATP provision. The mechanism of muscle contraction is elucidated through the sliding filament theory [17], where the myosin-heavy chain filament forms a cross-bridge with actin thin filament. Two demands need to be met for the contraction to occur. First, the muscle needs to be excited by motor neurones, resulting in sarcolemmal depolarisation and Ca^2+^ entry into the myoplasm which binds troponin C, relieving the troponin I block so that the cross-bridge can assemble. Second, ATP needs to be provided for the myosin head so that it does not remain immobilised with attached actin in a rigor state [18]. Importantly, slow-twitch oxidative (type-I) fibres hydrolyse ATP at a much slower rate than fast-twitch oxidative (type-IIA) and fast-twitch glycolytic (type-IIB) [19]. Therefore, type-I are deemed fatigue-resistant, and their abundance is attributed to a phenotype of endurance [11]. With a central decision to move/exercise faster and harder, the myoplasmic ATP pool becomes depleted [20], and therefore there is a call of duty for mitochondria to restore ATP homeostasis in the muscle through accelerated OXPHOS. Once OXPHOS meets the demand of the ATP hydrolysis at the myofilaments, a steady state is achieved as a new homeostatic level of oxidative metabolism is established. Consistently, enhanced kinetics and the ability to restore ATP through oxidative metabolism (rather than glycolytic) has been observed in professional endurance runners [21].

## 3. Oxidative Phosphorylation Is Supported by Reducing Agents Made in the TCA Cycle

OXPHOS is the process of ATP synthesis as a result of dissipating the potential of H^+^ gradient across the intermembrane (IMS) and matrix through mitochondrial complex V. The complexes I to IV utilise the reducing agents, NADH and reduced flavin adenine dinucleotide (FADH_2_) in concert, to pass electrons to O_2_, as they pump H^+^ into the IMS. NADH and FADH_2_ are synthesised in the mitochondrial matrix by the enzymes of the tricarboxylic acid (TCA) cycle [18]. Oxidative pathways of three main energy substrates (carbohydrates, fatty acids, and lactate) converge on the TCA cycle as acetyl-CoA. This point is critical for the determination of substrate preference, that is, between glycolytic (glucose and lactate) and fatty acid (FA). The events of switching between the carbon fuels are manifested systemically by the aforementioned critical intensities (LT or Fat Max). Partial contributions of the major substrates to the TCA cycle are inter-regulated on the basis of several factors, including substrate availability, sex differences, diet, training status, exercise duration, and most importantly on energy demands as the exercise intensity changes [22]. The ability to adapt the appropriate flux levels of carbohydrate and FA into the TCA cycle reflects the oxidative adaptation of endurance-trained individuals [23], while the contrasting inflexibility is synonymous with metabolic disorder and obesity [24]. 

## 4. Energy Demand Regulates Substrate Preference—Fat Is “Cheap”

Exercise intensity determines the contributions of carbohydrate and FA fluxes into the TCA cycle [23]. A mechanism of inhibitory cross-regulation between carbohydrate and fat mitochondrial metabolism was first reported by Randle et al. [25]. They observed a downregulated carbohydrate oxidation in artificially perfused cardiac muscle, due to increased fat availability, which has been explained to be a result of “glucose–fatty acid cycle”. Further years of research led to the emergence of the “crossover point” paradigm [26]. Accordingly, as the intensity of exercise increases beyond the previously described Fat Max [8], the carbohydrate oxidation abruptly rises to replace drastically falling FA flux (Figure 1). Brooks and Mercier suggest that endurance exercise training promotes adaptations in muscle energy metabolism reflected by improved fatty acid oxidation (FAO), concurrent with a decreased muscle sympathetic nervous system activity during moderate exercise or rest. Diminished pro-glycolytic, adrenergic stimulation further potentiates the enhancement of FAO. Regardless of training status, high-intensity exercise still requires a strong sympathetic stimulation to allow more powerful contractions and accelerated glycogenolysis. However, the onset of this increment parallel to FAO decline (crossover point) is delayed with consistent endurance training. Conclusively, the major resultant adaptation of skeletal muscles is the augmented activity of the parasympathetic nervous system to promote FA metabolism over carbohydrate, and muscle glycogen conservation (see later).

There is a trade-off between sustainability and power generation in exercise. For example, anaerobic glycolysis is more catalytically efficient than aerobic respiration, as it produces more ATP per protein mass, but it is less effective in restoring the ATP pool on a large scale compared with FA oxidation (FAO). Nilsson and colleagues [27] identified four optimal metabolic modes with decreasing substrate efficiency (ATP per cmol) but increasing catalytic capacity (ATP per [g of protein] per hour); (Figure 2). The most efficient is FAO, followed by glycogenolysis, NADH-dehydrogenase bypass, and fully anaerobic pathways that do not employ OXPHOS. Complex I deactivation has also been elucidated in carotid bodies in response to hypoxia [28,29]. As high-intensity muscle contractions can result in local PO_2_ to fall as low as 3–7 mmHg [30], it is plausible to consider a similar scenario to the carotid bodies, where hypoxia drives metabolic rewiring that ultimately promotes the chemosensory function [29] and where skeletal muscle’s function of generating high work rates may also be driven by hypoxia-modulated metabolism. Although the model of intermediary metabolism proposes that pathways exert cross-regulation on each other’s activities, the mechanism of the metabolic trade-off might be regulated centrally by the nervous system. That is, the elevation of metabolic rate and exercise-induced localised hypoxia may trigger a chemoreceptive response either through carotid bodies [31] or muscle type III and IV afferents [32,33], which causes an elevation in the muscle sympathetic nerve activity [34], resulting in increased plasma glucose transport [35] to support the need for glycolytic flux upregulation.

With increasing aerobic power output (*x*-axis) during an incremental exercise test to VO_2_ max, carbohydrate contribution (CHO%; *y*-axis) to oxidative metabolism increases exponentially until it reaches a near, but not whole, 100% (as lactate also contributes to oxidative metabolism). Fatty acid contribution (Fat%; *y*-axis) cannot fully satisfy OXPHOS (it can reach approximately 60%). Fat% function follows a reverse trend to CHO%, declining until FA contribution to OXPHOS is fully inhibited at near VO_2_ max work rates. The furthest FA and CHO oxidation rates crossover is 65% VO_2_ max. The model predicts that endurance training promotes a rightward shift of the crossover point resulting in elevated Fat Max and CHO sparing, while sympathetic stimulation induces larger CHO reliance at a given work rate.

Four optimal pathways (full orange circles) include the most efficient beta-oxidation through glycerol-3-phosphate shuttle and complex I bypass to fermentative glycolysis (does not enter TCA cycle, no electron transport chain (ETC); left in grey). The sub-optimal pathways (open orange circles) are malate-aspartate shuttle and hydrogen uncoupling (does not contribute to ATP synthesis). Glycolytic and lactate fluxes feed carbon and electrons into the TCA cycle (large) and NADH shuttles with fatty acid supply the alternative NADH pool (small). These energy flows are marked in blue if engaged in one of the optimal modes. Squares below the cycles represent the mitochondrial complexes 1–5 and electron-transferring flavoprotein dehydrogenase (ETF) as per the Figure 2A scheme. Complex I bypass shifts the curve right (dark blue), indicating higher substrate efficiency at high ATP synthesis/catalytic rates, relative to the alternative model (light blue).

Back to the steady state, skeletal muscles rely predominantly on intramuscular triglyceride (IMTG) stores, while actively contracting at aerobic work rate [36]. Endurance training has been shown to augment IMTG stores and enhance lipolytic capacity [37]. Endurance athletes have also been reported to contain higher levels of skeletal muscle lipid droplets (LDs) compared with untrained individuals. Paradoxically, LD content also strongly correlates with a degree of insulin resistance in type 2 diabetics, as Goodpaster et al. [38] reported a significant correlation (*r* = −0.57) between the histochemically quantified intramuscular lipid content and insulin sensitivity determined by the hyperinsulinemic-euglycemic clamp method. These conflicting findings led to years of debate concerning this “athlete’s paradox”. One could question: why do athletes share a similar trait with the antagonistic phenotype of metabolic inflexibility? The answer is rooted in the benefits of the oxidative phenotype. Beta oxidation provides the most ATP per carbon consumed; it is the most fuel-efficient energetic pathway [27]. Daemen et al. [39] resolve the paradox as they report significant differences in LD storage location and morphology between the two phenotypes. Type 2 diabetics contain large LDs mainly in the subsarcolemmal region of type II glycolytic fibres, while trained individuals store high numbers of small LDs in the intra-myofibrillar region of type I oxidative fibres, validating lipid storage as an adaptive trait of oxidative phenotype. Additionally, the LD programming process has been proven to be PGC-1α-dependent [40]. Peroxisome proliferator-activated receptor γ coactivator-1α (PGC-1α) is recognised as an essential regulator of mitochondrial biogenesis [41] by promoting the expression of mitochondrial transcription factor A (TFAM) [42]. Importantly, PGC-1α mRNA and protein content are upregulated with endurance exercise [43,44].

As the exercise intensity increases, the rate of fat oxidation rises until it reaches a peak between 40 and 65% of VO_2_ max, after which it falls precipitously. Although this trait is adaptable with training, around 65% of VO_2_ max seems to be the ultimate ceiling, even for endurance athletes [9,45]. Additionally, PFO and Fat Max weakly correlated with ultra-endurance performance in males [46] and females [47]. The above studies validate that FA flux cannot support the OXPHOS in exercise on its own, and despite being substrate-efficient, it tells only a part of the story of the oxidative phenotype. Interestingly, women have been considered better fat “metabolisers” as they have a higher relative whole-body fat oxidation at given steady-state exercise intensities compared with men [48]. Moreover, Fat Max in females occurs usually at higher fractions of VO_2_ max relative to males [49]. The most evident explanation for the difference lies in the ovarian hormone, oestrogen. Oestrogen-related receptor α (ERR-α) has been shown to promote mitochondrial biogenesis and transcription of target proteins through assembling a transcriptional factor with PGC-1α [50]. Of interest, post-menopausal women become exponentially more susceptible to hypertension [51], which is associated with aberrant endothelial reactive oxygen species (ROS) production [52], thus highlighting the significance of the evolution of the oxidative phenotype for overall REDOX homeostasis and health. Nonetheless, as FAO falls, the glycolytic flux via pyruvate (and lactate) supports OXPHOS increasingly until the Fat Min intensity is reached (around 80% of VO_2_ max) [9], beyond which the anaerobic metabolism predominates. The fact that endurance training can improve lipolysis, fat oxidation capacity, and Fat Max [53,54,55,56] is not only significant from the point of efficiency but also the conservation of endogenous carbohydrate stores. If the contributions of IMTG and free fatty acids (FFA) are critically diminished, the exercise continuation relies on intramuscular glycogen and blood glucose (and lactate to a small degree). However, given that the fully anaerobic zone (beyond the onset of blood lactate accumulation) is not entered, this scenario in between Fat Max and OBLA is an unsustainable model of exercise because human carbohydrate storage is far from being infinite and for most does not exceed 3000 kcal (740 g) [57]. Around 80% is stored in skeletal muscles and 10–15% in the liver [58]. Depletion of the intramuscular glycogen has been long implicated in contractile dysfunction and fatigue, and it is recognised among marathon runners as “hitting the wall” [59]. It has been speculated that the endogenous glycogen location is critical for contractile function with intra-myofibrillar stores being the main players [60]. De facto, this trait is malleable as 2 weeks of cast immobilisation resulted in a 50% fall of intra-myofibrillar glycogen in *vastus lateralis* [61]. On the other side of the spectrum, elite endurance athletes were reported to have unusually large intra-myofibrillar pools [61], and these are primarily found in type-I fibres where restoration can be achieved in the recovery period after only one bout of exercise [62]. Regarding the mechanism of fatigue due to glycogen shortage, intra-myofibrillar depletion strongly correlated with the reduction in tetanus-induced SR Ca^2+^ release [63]. More recently, glycogen depletion in ex vivo fibres have been shown to attenuate Na^+^/K^+^-ATPase activity, which in turn increases the re-priming period (interval between action potentials) and reduces force, despite high global non-glycogenetic ATP levels [64,65]. Nevertheless, Fat Max occurs at a relatively low fraction (40–65%) of VO_2_ max [45]; therefore, in the context of most endurance events, this intensity has to be breached for optimum performance. The glycolytic flux has to be employed, and considering finite sources of muscle glycogen, in-race feeding [66] as well as pre-race carbohydrate “loading” strategies are commonly and necessarily employed, although the latter remains controversial [67].

## 5. Balancing between Fatty Acids and Pyruvate to Meet the Contractile Demand

A crucial point in the FA oxidation pathway is FA transport across mitochondrial membranes. The process is facilitated by carnitine palmitoyl transferases (CPT-1 and -2), carnitine acylcarnitine translocase (CACT), and the critical shuttle molecule carnitine Figure 3. A FA synthesis precursor, malonyl-CoA, has been long been identified to exert direct inhibition on outer-membrane CPT-1. Rat skeletal muscle malonyl-CoA concentration has been reported to fall following 30 min of wheel running. Consistent with the crossover paradigm [26] and the observation of a profound drop in the FAO rate beyond Fat Max [8,9], an inhibitory point somewhere along the FA flux has to be activated through increasing glycolytic flux, ROS generation, or depolarisation and SR Ca^2+^ release. There are six points along the FAO process where the inhibition can be exerted, either on transport to skeletal muscles or directly on oxidative enzymes [23]. First is FFA transport across the muscle fibre membrane. FA-binding proteins and transporters such as FAT/CD36 have been shown to translocate to the sarcolemma after moderate exercise [68,69]. This translocation has not been directly tested in high-intensity exercise, but an effort at 80% of VO_2_ max has been reported to specifically downregulate oxidation of transport-relying long-chain fatty acid (LCFA), but not of medium-chain FA (MCFA) [70]. However, this effect could be facilitated by mitochondrial transporter regulation, especially considering that IMTG is considered a predominant FA source for β-oxidation over FFA [71]. Although CD36 does not appear to be essential, it has been identified as an enhancing trait of FAO with endurance training, as regular exercise significantly elevated its expression, which correlated with the boost in FAO rates [72,73]. Notably, CD36 protein levels remained upregulated for 3 days following a single bout of 45 min of cycling [72], highlighting the persistence of the training oxidative effect into a prolonged advantageous phenotype. Moreover, FAT/CD36 is transcribed 85% more in women than men, following a 90-min cycle at 60% of VO_2_ max. Additionally, CD36 protein levels are nearly 50% greater in females, irrespective of training status [74]. Therefore, this facilitatory transporter that co-precipitates with a crucial mitochondrial transporter CPT-1 in skeletal muscles [72] partially explains the oxidative prowess of pre-menopausal females. Cytoplasmic FFA transport and IMTG synthesis do not appear to be limiting factors of FAO during exercise. Exercise-induced elevation in circulating catecholamines stimulates muscle and white adipose tissue (WAT) lipolysis, raising the plasma FFA concentrations [71]. Interestingly, there is a thin line between adaptive and maladaptive outcome when it comes to WAT lipolysis and FFA transport in response to exercise. Namely, poor circulation to WAT in type-2 diabetics impedes epinephrine from inducing lipolysis, therefore contributing to exercise intolerance in the dyslipidemic phenotype [71]. Inhibition of hydrolysis and oxidation of IMTG is a possible limitation of FAO. The adipose triglyceride lipase (ATGL) and hormone-sensitive lipase (HSL) are the enzymes that mediate IMTG hydrolysis [75]. Perilipins that coat LDs separate IMTG from, and hence inhibit, TG lipases. Moderate exercise results in adrenergically induced phosphorylation of HSL and ATGL, as well as 5’ adenosine monophosphate-activated protein kinase (AMPK)-mediated perilipin phosphorylation, subsequent LD and enzyme assembly, and augmented IMTG hydrolysis [76]. HSL activity is optimal at pH 7.0, and its lipolytic rates increase with exercise [77]. Moreover, 8 weeks of wheel running resulted in elevated expression of HSL, ATGL, and another lipase CGI-58, as well as significantly increased perilipin-5 phosphorylation ratio in obese mice [53]. It has been speculated that, at higher work rates causing a larger disturbance in ATP homeostasis, AMPK hyper-phosphorylates HSL resulting in lower catalytic activity [78]. Another function of AMPK in the regulation of FAO through modulating FA mitochondrial transport has been proposed by Richter and Ruderman [79]. Namely, with increasing muscle ATP dyshomeostasis, AMPK activates acetyl-CoA carboxylase (ACC). ACC converts acetyl-CoA to malonyl-CoA, which is a main CPT-1 allosteric inhibitor (as is discussed later). This is consistent with the observation of reduced respiratory exchange ratio (RER) during wheel running in AMPK β1β2 isoform null mice [80]. However, a clear-cut role of AMPK in regulating FAO remains elusive, as AMPK α1 knockout mice were shown to not exhibit altered fat metabolism during treadmill running [81]. Nonetheless, all of the lipolytic and FA transport pathways converge on β-oxidation inside the matrix. The key enzyme, β-hydroxy acyl-CoA dehydrogenase (HAD), removes H^+^ from FA, producing acetyl-CoA groups. There is a lack of evidence supporting the notion that the catalytic activity of β-oxidation enzymes is limiting to FAO. Multiple studies showed that long-chain FAO specifically is depressed with higher glycolytic flux, indicating the main limitation is in the FA transport system [82]. Nonetheless, β-oxidation catalytic capacity can be enhanced by an endurance (but not resistance) training regime, which significantly elevated intramuscular HAD content [83]. A crucial point in the FA oxidation pathway is the FA transport across mitochondrial membranes. The process is facilitated by carnitine palmitoyl transferases (CPT-1 and -2), carnitine acylcarnitine translocase (CACT), and the critical shuttle molecule carnitine (Figure 3). A FA synthesis precursor, malonyl-CoA, has long been identified to exert direct inhibition on outer-membrane CPT-1. Rat skeletal muscle malonyl-CoA concentration has been reported to fall following 30 min of wheel running [84]. However, studies in human have shown that there are no significant changes in malonyl-CoA after bouts of exercise at intensities from 35% to 100% of VO_2_ max [85,86,87], thus suggesting that the exercise-dependent mechanism of FAO limitation is distinct to the biosynthetic one. Another possibility for the abrupt fall in FAO at higher work rates might be lactic acidosis, as Starrit et al. [88] reported a significant decrement in CPT-1 activity at a pH of 6.8 in isolated mitochondria. Moreover, FAT/CD36 specific inhibitor decreased palmitate oxidation by 95% [89], implying that mitochondrial outer membrane CD36 might play a role in enhancement [72,73], as well as negative regulation of FAO in exercise. Bezaire et al. [89] also showed that products of high ATP turnover (ADP, AMP, and inorganic phosphate) or high intracellular Ca^2+^ concentration do not exert any inhibitory effect on the CPT-1 activity in isolated mitochondria. This again suggests that the dramatic fall in FAO needs to involve intermediary metabolites that could link the major energy fluxes. Mitochondrial carnitine emerged as this intermediary agent, responsible for the regulation of the FA and glycolytic fluxes [82].

FA acyl-CoA–carnitine shuttle describes the role of carnitine and FA mitochondrial membrane transport at low-to-moderate exercise intensities. With the work rate approaching Fat Max, acetyl-CoA is extensively produced from pyruvate oxidation in the glycolytic pathway. This reaction is catalysed by the pyruvate dehydrogenase complex (PDC) located inside the matrix. Consequently, increasing matrix acetyl-CoA might be buffered by free carnitine to form acetyl-carnitine, thus “stealing” the critical metabolite of the FA transport machinery. On the basis of this paradigm, researchers have speculated that carnitine is the main FAO regulatory link [82,90]. No studies to date have discerned if carnitine mitochondrial levels are adjustable with training, nor what concentrations are critical to limit FAO. Notably, free carnitine levels are significantly lower following endurance exercise (and to a lesser extent sprint training) than resting ones [91]. Moreover, it is not elucidated how low the muscle-free carnitine needs to get during exercise to “bottleneck” the LCFA transport. Arenas and colleagues [91] reported elevated muscle carnitine in athletes under endogenous l-carnitine supplementation. However, other groups remained sceptical about the effectiveness of such supplementation [92]. Another study showed that 2 weeks of l-carnitine administration did not affect the FAO rate in a moderate 1.5-h cycling exercise [93,94]. Broad and associates did observe significantly declined plasma glucose during aerobic exercise in the l-carnitine supplemented group in contrast to controls after a high-fat dietary intake, supporting the role of carnitine in manipulating the fluxes. Moreover, 6 months of supplementation were shown to increase muscle total carnitine by 21% [95]. The same study reported over 50% glycogen sparing and 31% lower PDC activity following a 30-min cycle at 50% of VO_2_ max relative to controls. These findings support the hypothesis of free carnitine availability being limiting to FAO in exercise [90]. Strikingly, however, the same l-carnitine-supplemented individuals exhibited 38% higher PDC activation, along with 44% lower muscle lactate and smaller disturbance in PCr/ATP ratio, suggesting that carnitine does not only support (may also limit) the FAO but additionally enhances the glycolytic flux through PDC and better matching of OXPHOS to high (80% of VO_2_ max) energetic demand by better maintenance of the immediate ATP pool [13]. Additionally, 6-month l-carnitine supplementation led to an 11% improvement relative to controls in an all-out 30-min cycling time trial [95]. Ergo, one could argue that mitochondrial carnitine is passed between the glycolytic and FA fluxes to fine-tune the metabolic output in response to contractile demand. It is also suggested that CrAT ameliorates the rise of steady state and the consequent O_2_-deficit or rather the metabolic inertia [16].

## 6. Carnitine Acetyltransferase Mitigates Metabolic Inertia—A Trait of Oxidative Flexibility

The attainment of steady state in exercise has been historically attributed to delayed O_2_ delivery to muscles [12]. A more recent view includes delayed enzyme activation and carbon flux into the oxidative machinery of the TCA cycle and ultimately OXPHOS, that is, metabolic inertia [16]. Acetyl-CoA, which is the bridging point of oxidative metabolism, has a central role. Considering the observations implicating mitochondrial carnitine in the control of both FA and glycolytic fluxes [95], the light of attention is shone at carnitine acetyltransferase (CrAT) enzyme, which has previously been speculated to mediate the beneficial effects of l-carnitine supplementation in obese mice [96]. Consistently, CrAT activity has been shown to be significantly impeded in obese and diabetic rodents [97]. Bearing in mind the high tissue-specific expression of CrAT in skeletal and cardiac muscles [96], CrAT emerges as the critical player in endurance adaptation and performance. What makes CrAT even more special is its ability to drive a forward acylcarnitine synthesis and reverse carnitine and acetyl-CoA regeneration with an equilibrium constant of 1.5 implicating high reversibility [98]. Therefore, a forward CrAT reaction may potentially limit FAO by “stealing” free carnitines. Moreover, as excess acetyl-CoA activates pyruvate dehydrogenase kinases (PDK) 1–4, phosphorylating and inactivating E1α PDC subunit [99], the forward CrAT reaction may enhance the glycolytic flux. The inverse relationship of falling carnitine and increasing acylcarnitine with increments in exercise across 40–100% VO_2_ max range [100] would imply the acceleration of the forward CrAT activity, consistent with glycolytic flux being more suitable in matching the high-end oxidative demands [27]. Transcending evidence for CrAT being central to a powerful oxidative phenotype comes from experiments involving CrAT-deficient animals. First of all, muscle-specific CrAT null mice displayed most features of the metabolic disorder as they quickly gained weight, were glucose intolerant, and were insulin resistant [101]. Enigmatically, these animals had lower levels of serum FFA, which is a beneficial effect, although this is most likely a result of relieved FA acyl-CoA carnitine shuttle inhibition as evidenced by increased palmitate and oleate fed oxidation in perfused fibres [101] and decreased RER in exercise [102]. Interestingly, myotubes transfected with adenoviruses engineered to silence CrAT (rAd-shCrat) presented enhanced oleate oxidation and slightly, but significantly, depressed glucose uptake [101]. Correspondingly, virally induced CrAT overexpression (rAd-Crat) resulted in a profound increase in acylcarnitine efflux and PDC activity. These observations propose crucial mechanisms that are normally very flexible in the oxidative phenotype of an endurance athlete but are dysregulated in the context of obesity and metabolic syndrome [102]. Consistent with these observations and its efficacy in improving endurance performance [91,95], l-carnitine supplementation has been employed in clinical trials to treat metabolic syndrome. A very recent meta-analysis summarises the success of this approach with significant reductions of fasting plasma glucose, insulin, homeostatic model assessment for insulin resistance (HOMA-IR), and glycosylated haemoglobin (HbA1C) concentrations [103].

CrAT activity is not fixed in one direction (e.g., acylcarnitine efflux or carnitine recycling), but is continuously shifting the equilibrium in order to facilitate both FA and glycolytic fluxes. It is plausible that with the rise of metabolic demand above given steady state, CrAT recovers acetyl-CoAs from acylcarnitine to support the “starved” TCA cycle, but when the glycolytic flux machinery is ready to meet the demand, CrAT mops up excess acetyl groups with carnitine [102]. Hence, supplementary solutions seem rather uni-directional. What we need for a functional oxidative phenotype is a fully flexible metabolism, by virtue of highly “responsive” CrAT. Loss of CrAT function might be central to metabolic disease considering that OXPHOS lag is synonymous with O_2_ deficit [16], and that muscle type III and IV afferents, which have been shown to promote central fatigue [104,105] and to limit top-end exercise performance [32,33], might fire prematurely due to metabolic inflexibility in obese individuals. This may consequently cause the sensation of dyspnoea and fatigue, leading to the cessation of exercise and a vicious cycle of physical inactivity. Indeed, Seiler et al. [102] report that skeletal muscle-specific CrAT knockout mice fatigue faster in an incremental trail of treadmill running, compared to controls. Interestingly, the impact of CrAT on whole-body metabolic proficiency appears to be stepping out beyond skeletal muscle.

Elevated circulating branch-chain amino acids (BCAA) have been implicated in mediating insulin resistance in obesity [106], while gastric bypass surgeries which promote profound weight loss were shown to decrease blood BCAA [107]. BCAA can be utilised in de novo FA synthesis to generate monomethyl branched-chain FA (mmBCFA) [108]. Consistent with the finding of the depressed catabolic capacity of BCAA in obese individuals [109], total adipose tissue mmBCFA content was 30% lower in obese than in lean subjects and increased by 65% after gastric bypass surgery. Moreover, blood mmBCFA concentrations correlated positively with responsiveness to insulin [110]. The mmBCFA catalytic process has been shown to be mediated by “promiscuous” CrAT activity in WAT. This biosynthetic pathway was remarkably inhibited by local WAT hypoxia in obese animals [111].

Additionally, acetyl-CoA buffering through CrAT at rest (steady state) was observed to prevent the macronutrient load-induced post-translational lysine acetylation of mitochondrial proteins, which is synonymous with glucose intolerance and elevated insulin resistance [112]. These findings are consistent with previous reports of SIRT3 deacetylase-deficient mice developing metabolic syndrome [113] and ventricular dysfunction [114].

The significance of CrAT for the maintenance of the oxidative phenotype may extend to the neural regulation of energy homeostasis. Hypothalamic, agouti-related protein (AgRP)-secreting neurons increase food intake [115]. AgRP-specific CrAT knockout mice were found to have increased systemic FA oxidation levels, with declined ability to switch to glucose utilisation after refeeding [116], mirroring the effect of muscle-specific knockouts which were glucose-intolerant [101]. Furthermore, CrAT-deficient mice were observed to eat significantly more following a calorie restriction diet resulting in a significant rebound weight gain [116]. Bearing in mind the abundant evidence for PGC-1α-regulated CrAT expression [102] to be essential for metabolic proficiency and endurance exercise performance, further research to design best-suited training strategies is warranted. However, CrAT-coordinated muscle metabolism cannot explain the entire oxidative capacity, as CrAT null mice did not show any discrepancy compared with controls in blood accumulation of lactate during incremental treadmill running test [102]. Historically, lactate was believed to be a waste product of the anaerobic metabolism of pyruvate, but this has been reconsidered with the birth of the “lactate shuttle” theory [117]. These days, lactate is considered to support OXPHOS in fully aerobic and accelerated metabolism, with the extreme energy demand-induced accumulation improving the metabolic capacity through adaptive signalling [118].

## 7. Lactate Supports the Glycolytic Flux through Oxidation in Mitochondria

In terms of exercise, lactate is shuttled between the glycolytic and oxidative fibres and between skeletal muscles as a whole, liver, and heart. Additionally, the intracellular shuttles include those between cytosol mitochondria [119] and cytosol peroxisome [120]. These exchanges, driven by the lactate concentration gradient or REDOX state, are facilitated by monocarboxylate transporters (MCT), with 1–4 being the most well-characterised. Three muscle fibre types express differential content of MCT1–4 [121,122]. Type I oxidative express predominantly MCT1, with a higher affinity for lactate, which facilitates the uptake, while the type IIb glycolytic fibres mostly express MCT4, which usually promotes lactate efflux. The net outcome of inter-fibre shuttling of lactate reflects the exercise intensity. At a relatively moderate work rate, the rate of appearance in the blood is matched by the rate of disposal [123]. However, beyond LT intensity, the accumulation exceeds the disposal, resulting in elevated blood [La^−^]. This is consistent with the size principle of motor recruitment [124]. The small units abundant in oxidative fibres are recruited first, but with increased neural stimulation and demand to generate higher work rates, larger and more powerful glycolytic units are recruited. Therefore, more lactate efflux occurs, which is facilitated by pyruvate to lactate reduction, by lactate dehydrogenase A (LDHA) associated with glycolytic fibres [125]. On the contrary, LDHB mediates lactate oxidation to pyruvate in the recycling type I fibres. Notably, there is no clear-cut evidence that glycolysis terminates on pyruvate; the more recent view is that lactate is the terminal intermediate even in fully aerobic metabolism [126]. Because blood lactate accumulation correlates with increasing work rate and VO_2_ in the high end of the incremental exercise test, lactic acidosis has been speculated to interrupt contractile function and lead to fatigue. However, this concept has been disproved [127]. Notably, pyruvate reduction should not cause acidosis as lactate anions and hydrogen protons are generated in a 1:1 ratio [128]. Lactate supplementation has been reported to improve 90-min endurance performance [129] and cardiac contractile function in heart failure patients [130]. A more popular concept describes lactate as a fatigue-protector as it prevents the negative effects of high interstitial potassium concentration on muscle excitability [131].

Nevertheless, a major role of lactate appears to be its energetic value in oxidative metabolism, as it has been demonstrated to support OXPHOS just as efficiently as pyruvate [132]. Their assessment of lactate-fed ROS generation in permeabilised mitochondria suggests the presence of mitochondrial lactate dehydrogenase, which is consistent with many previous studies summarised in a 2014 review by Passarella et al. [133]. Passarella and colleagues do acknowledge a strong opposing front of the debate. A more recent study supports the counter notion, as neither in skeletal muscle nor cardiomyocyte-isolated mitochondria does lactate match the energetic efficiency of pyruvate [134]. Nonetheless, one important aspect that Young and colleagues [132] seem to neglect in their discussion is a significantly lower ability of lactate to support ADP-induced state 3 respiration in muscle mitochondria, which should have major implications in high-intensity exercise and the ability to recycle lactate. The main argument against lactate oxidation is a high NADH/NAD^+^ ratio inside the mitochondrial matrix [135]. However, what these studies did not take into account is a differential NADH handling with different OXPHOS “velocities”. Consistent with the recent mitochondrial metabolism model [27], aspartate-malate and glycerol-3-phosphate shuttles collaborate to feed NADH-derived electrons directly into ubiquinone (QH_2_), and thus they enter the electron transport chain at complex III, with NADH dehydrogenase (complex I) being bypassed, therefore resulting in a significant NADH accumulation downstream in the matrix [29]. As these alternative shuttles are engaged toward higher catalytic capacity/energy demand, increasing complex I bypass and NADH accumulation should result in lactate being progressively less oxidised in mitochondria, resulting in the imbalance between accumulation versus disposal rates and subsequent exponential rise in blood [La^−^]. This prediction is consistent with the skeletal muscle adaptation to prolonged high altitude, where malate-aspartate and glycerol-3-phosphate shuttle enzymes expression is significantly elevated, resulting in an increased resting blood [La^−^] [136]. Therefore, the reason skeletal muscle cannot oxidise lactate as efficiently as pyruvate is due to its ability to accelerate the OXPHOS to extreme levels, which is exactly the opposite of non-fatigable cardiac muscle (which does not generate high metabolic demand).

A recent study supports the notion of lactate being an important energy source for the generation of high oxidative power during the incremental VO_2_ max protocol [137]. The blood lactate accumulation profile trended nearly in a perfect match with the carbohydrate oxidation (CHOox) rate in untrained subjects during a cycling test to VO_2_ max, whereas elite cyclists tended to maintain blood lactate more stable for longer until it began to reflect the CHOox rate. Interestingly, in patients with metabolic syndrome, the blood lactate curve exceeded CHOox (relative to power output) early on in the trial, enforcing the concept of lactate accumulation as a reflection of overwhelmed glycolytic flux into the oxidative metabolism. Moreover, a robust inverse relationship between [La^−^] and FA oxidation rate was observed across all levels of fitness, which considering [La^−^] nearly mirroring CHOox at a given power is consistent with the early crossover concept [26]. These exciting findings are supported by the earlier work from George Brooks’ lab, as they showed that lactate flux into oxidative metabolism can exceed the glucose flux four times during cycling exercise at 65% of VO_2_ max [138,139].

Therefore, the critical feature of a powerful oxidative phenotype is the employment of high levels of FAO to preserve glycogen, as well as high lactate oxidative capacity to also, in a sense, preserve glycogen by recycling the glycolysis-derived carbons through lactate shuttling, which accelerates with increased energetic demand. This is consistent with the observation of PGC-1α-transfected myotubes overexpressing LDHB and MCT1 facilitating improved oxidation and decreased accumulation of lactate [140]. Another ex vivo study showed that an acute addition of lactic acid inhibited glucose and FA oxidation but improved FA uptake (counteracting ROS-dependent lipid peroxidation) in myotubes [141]. Thus, one could speculate that lactate ultimately halts the oxidative machinery from inducing too large of an oxidative insult, and hence the lactate correlation (but not proven causality) with fatigue at high-end intensities above 75% of VO_2_ max. This is not, however, where the story of the oxidative phenotype ends. Indeed, it is just beginning, with the stage set for adaptive signalling.

## 8. Lactate Sets the Stage for PGC-1α Signalling—The Oxidative Booster

If lactate’s function is to trigger an adaptive response to disruption in ATP homeostasis during high-intensity exercise, it should facilitate a more efficient oxidative system that will hopefully meet similar energy demand on the next occasion without a need to employ the energetic, yet critical, power of lactate. Interestingly, high-altitude sojourns have been associated with a progressively regressing VO_2_ max [142] as well as declining lactate accumulation at a given work rate [143]. Together with a marked fall in muscle mitochondrial density [144], one can draw a connection between lactate stimulus and mitochondrial biogenesis and capacity. A transcriptional co-activator PGC-1α is central to the development of these oxidative traits. PGC-1α has been demonstrated to promote fibre switching to oxidative type-I and upregulation of electron transport chain constituents including cytochrome c, cytochrome oxidase (COX), and ATP synthase [145,146,147]. Additionally, PGC-1α-overexpressing mice presented an elevated expression of citrate synthase (CS), a primary enzyme of the TCA cycle joining acetyl-CoA with 4-carbon oxaloacetate, which was additionally reflected in the study by improved VO_2_ max [147]. In vitro studies have proven PGC-1α-coordinated enhancement of fat metabolism through increased expression of CPT-1 and a β-oxidation enzyme, medium-chain acyl-CoA dehydrogenase (MCAD) [148]. Similar expression patterns were observed in vivo in PGC-1α transgenic mice, with additional observation of increased long-chain acyl-CoA dehydrogenase (LCAD), very-long-chain acyl dehydrogenase (ACADVL), FAT/CD36, FA binding protein 3 (FABP3), and FA transport protein 1 (FATP1) [146,147]. It is evident that PGC-1α mediates oxidative development by inducing an overall mitochondrial “growth spurt” and enhanced FA side of carbon flux into the TCA cycle. The upregulation of glycolytic flux would be an intuitive outcome in order to maximise the VO_2_ max. However, the PGC-1α effects on this side of carbon flow appear biphasic, with an improved glucose sarcolemma uptake, but somewhat declined, or rather “more tightly regulated” mitochondrial component of glycolytic flux. Insulin-regulated glucose transporter GLUT4 abundance has been reported to be upregulated in a PGC-1α-dependent manner in vitro [149]. Moreover, muscle GLUT4 expression declined in PGC-1α knockout animals [146]. Wende and colleagues also reported that PGC-1α-transfected animals presented elevated levels of hexokinase II (HKII) protein, which phosphorylates and therefore traps glucose intracellularly in the investment phase of glycolysis. Contrastingly, a downstream glycolytic enzyme, phosphofructokinase (PFK), was downregulated in the same transgenic animals, suggesting a tendency for the accumulation of glucose rather than its immediate oxidation [146]. This is consistent with significantly lower glycogen phosphorylase mRNA in PGC-1α-overexpressing animals, implying reduced glycogenolysis at rest [147]. Apart from having significantly larger intramuscular glycogen stores relative to wild type, these mutant mice present an enormous ability to preserve glycogen with prolonged, as well as high-intensity treadmill running [146]. These findings are consistent with the crossover concept, whereby regular endurance exercise promotes a rightward shift of the Fat Max point, promoting the sparing of limited endogenous carbohydrate [26]. Aerobic exercise-induced PGC-1α signal also establishes a more “competent” regulation of the glycolytic flux by boosting the expression of PDK4 [146,147] a negative regulator of PDC with the second-highest affinity for E1α subunit, among PDK isoforms [99]. PDK1 expression has also been reported to be upregulated by PPARδ [150]. The PPAR family of nuclear receptors can be activated by an elevated content of dietary FAs [151]. This is consistent with the early observations of the abundant substrate preference in the “glucose–fatty acid cycle” [25]. Therefore, one could speculate that a high-fat dietary intervention such as the popular high-fat, very low-carbohydrate diet should accentuate the exercise effect on oxidative capacity and metabolic efficiency. Ketogenic diet interventions elevated LT in off-road advanced cyclists [152] and attenuated glycogen depletion in steady-state exercise at 75% of VO_2_ max in ultra-endurance runners [153]. Interestingly, keto-adapted runners had significantly higher pre- and post-exercise free carnitine levels, which could suggest enhanced metabolic flexibility [102]. Additionally, these ketogenic athletes presented nearly eight times larger acylcarnitine content as carbohydrate-diet controls pre-exercise, which intricately has been associated with obesity and CrAT dysfunction in animal models [97]. However, these runners had an astounding ability to consume these elevated levels in exercise to comparably the same levels as controls, indicating not only special CrAT efficiency but also elevated acetyl-CoA levels as per reverse CrAT reaction [102]. These observations in ketogenic athletes would mean an improved PDC negative regulation in exercise due to high acetyl-CoA/CoA ratio [25] due to PDK4 activation [99]. Consequently, these observations support the glycogen preservation paradigm.

However, a major question remains: how does the high-fat diet not induce CrAT malfunction with long-chain FA acyl-CoA excess as seen with obese animals [97]? A high-fat diet has been shown to induce inflammation in the hypothalamic melanocortin system in mice and the AgRP orexigenic neurons in vitro [154]. As mentioned previously, AgRP-specific CrAT knockout animals were shown to have severe metabolic inflexibility, characteristic of type II diabetics and obesity [116]. This culprit effect on the hypothalamic feeding centre was profoundly exacerbated in PGC-1α knockouts, resulting in depressed ERR-α activation and localised insulin resistance [155]. These studies suggest that PGC-1α promotes an oxidative phenotype and metabolic homeostasis also centrally. Chronic WAT inflammation, implicated in obesity, has been shown to translate to muscle-specific downregulation of PGC-1α in genetically (leptin-deficient) and diet-induced obese mice [156]. Ultimately, it is the oxidative capacity of mitochondria that manages the macronutrient energetic overload and prevents the maladaptive effects, as shown in TFAM transgenic mice [157]. When viewed together, these studies draw an arc of metabolic proficiency between the central homeostatic regulator, main peripheral energy consumer (skeletal muscles), and the main peripheral energy store (adipose tissue).

Drawing inferences from the above studies, we can establish an indisputable link between the elevated work rate, blood lactate accumulation, PGC-1α signalling, improved lactate clearance and downregulated efflux [140], enhanced FA oxidative capacity and ability to mitigate metabolic inertia, and the ability to efficiently organise the storage of macronutrient energy and the ultimate feature of endurance phenotype, namely, the endogenous glycogen conservation. However, the lactate signal is not enough to satisfy all of the adaptations. Direct lactate signalling has been reported to inhibit WAT lipolysis through G-protein-coupled receptor 81, expressed in adipocytes [158,159]. Furthermore, the mean lactate concentration to elicit half of the maximal response (EC_50_) was 4.87 mmol/L [159], which is of physiological significance in terms of exercise with the OBLA achieving around 4 mmol/L on average [6]. Additionally, lactate has been implicated in promoting O_2_-sensing in carotid bodies, as lactate receptor Olfr78-deficient mice fail to respond to hypoxia, but not hypercapnia [160]. Even GRPR81 have been described in multiple sites of the brain, where they promote angiogenesis via activation of vascular endothelial growth factor A (VEGFA) [161]. However, no such receptors have been identified in skeletal muscle fibres to date. A recent study seems to clear the clouds of mystery, as an antioxidant treatment with *N*-acetyl-l-cysteine (NAC) prevented lactate-mediated induction of PGC-1α upregulation in myotubes [162]. This passes the baton to ROS as the main protagonist on the stage of oxidative adaptation.

## 9. Reactive Oxygen Species—A Thin Line between Fit and Failed

ROS and reactive nitrogen species (RNS) generation in muscle during exercise is a widely accepted phenomenon. These molecules contain an unpaired electron which makes them extremely reactive radicals. These include superoxide (O_2_^−^), nitric oxide (NO), hydroxyl radical (OH^−^), and hydrogen peroxide (H_2_O_2_) [163,164]. However, there is still a disagreement as to what are the main sources of ROS in exercise. The potential sites are mitochondria (electron leak from ETC), NADPH oxidases (NOX2, NOX4, DUOX1, DUOX4), phospholipase A_2_ (PLA_2_), xanthine oxidase (XO), and lipoxygenases [165]. Additionally, a high-power output exercise can lead to the tumour necrosis factor (TNF)-α and interleukin-6 (IL-6)-induced neutrophil and macrophage activity. These phagocytic immune cells use NOX2 for “oxidative burst”, that is, ROS release into the inter-myofibrillar space, likely contributing to the post-exercise inflammation and soreness [166]. Moreover, the exercise-associated catecholamine release promotes endothelium-derived XO [167]. An important question to address is how could the muscle ROS production—which has been implicated in myo-pathologies like ageing-related sarcopenia [168], Duchene muscular dystrophy (DMD) [169], and diaphragm dysfunction in respiratory diseases [170]—possibly benefit the oxidative adaptation to exercise? The answer is coded in the biphasic effects of ROS on contractile function and adaptation. The proposed model predicts that increased ROS production facilitates the force-generating capacity to a peak point, after which excessive oxidisation of the contractile machinery results in a precipitous decline in force generation [171]. A potential mechanism elucidated involves alteration of troponin C affinity to intracellular Ca^2+^ and myosin–actin binding or affecting muscle excitability and SR Ca^2+^ release [172]. On the adaptive flipside, ROS are essential for signalling, as multiple studies including exogenous antioxidant treatments were shown to inhibit training effects in skeletal muscle [173]. However, chronic physical stress may lead to overtraining syndrome, characterised by excess ROS generation due to incompetent levels of endogenous antioxidant capacity, resulting in a marked decline in exercise performance [174,175]. Prolonged elevation in myocellular ROS can result in sustained activation of the nuclear factor kappa-light-chain-enhancer of activated B (NF-κB) and forkhead box (FOXO), which in turn activate atrogin-1, muscle atrophy F-box (MAFbx), and muscle RING finger protein 1 (MuRF-1) [176]. MAFbx and MuRF-1 then target myofibrillar proteins, including myosin light and heavy chains for degradation [177]. If consistent exercise training is complemented with adequate recovery, ROS-mediated adaptation can occur, including upregulation of antioxidant enzymes, namely, superoxide dismutase (SOD), glutathione peroxidase (GPX), and catalase [164]. Similar to the paradigm of metabolic stress leading to improvement of metabolic flexibility to avoid future ATP dyshomeostasis, oxidative stress leads to antioxidant system adaptation to avoid the REDOX imbalance when exposed to similar stress again. The antioxidant adaptation is mediated by the aforementioned PGC-1α, whose overexpression resulted in increased antioxidant SOD2 and GPX1 mRNA, and reduction in ROS accumulation, paralleled by the diminished electron leak and elevated intermembrane space potential [178,179,180]. However, the inflammatory response and oxidative stress are often considered to go hand in hand, and several studies have suggested that the appearance of inflammatory and oxidative stress markers are independent of each other, as vigorous walking increased IL-6 and TNF-α but not markers of oxidative stress immediately after exercise [181]. Similarly, inflammatory cytokines but not markers of DNA oxidation were elevated in runners in the minutes after completing a marathon [182]. On the contrary, DNA oxidation markers were significantly increased immediately after and for another 90 min in half-marathon competitors [183]. Moreover, Jamurtas et al. [184] reported that only 4 × 30 s all-out cycling protocol is sufficient to elevate protein carbonylation in comparison to baseline. Notably, the same study exemplified the supremacy of short bouts of high-intensity relative to continuous 30 min of cycling at 70% of VO_2_ max in terms of inducing the immune response and oxidative stress. Notwithstanding the potential benefits, high-intensity interval training (HIIT) should be avoided or carefully supplemented to the moderate endurance exercise training regime in diseased individuals, in this way ameliorating the compromised phenotype while minimising the possibility of diminishing returns due to acute oxidative stress on top of chronic stress. This notion was supported by the studies where in contrast to beneficial moderate endurance training, HIIT exacerbated cardiovascular disease in hypertensive rats [185,186].

Endurance exercise-induced PGC-1α signalling may promote indirect antioxidant functions in mitochondria, as PGC-1α has been recognised to orchestrate brown adipose tissue thermogenesis through uncoupling protein (UCP) 1 [187], which dissipates the proton gradient across intermembrane space. Considering, that the electron leak at NADH dehydrogenase is particularly sensitive to mild uncoupling [188], reducing the membrane potential across the inner mitochondrial membrane might be the first front of the antioxidant defence system. Additionally, PGC-1α has been shown to activate SIRT3, which in turn deacetylases SOD2, enhancing its antioxidant activity [189]. SIRT3 promotes energy homeostasis by directly upregulating the activity of NADH dehydrogenase [190] and succinate dehydrogenase [191]. Considering that SIRT3 knockout mice develop metabolic syndrome [113], the above studies highlight the multi-functionality of PGC-1α effector pathways by regulating the energy and REDOX homeostasis in unison.

When it comes to potential mediators of ROS to PGC-1α signal, p38 mitogen-activated protein kinase (MAPK), NF-κB, and Ca^2+^-dependent calmodulin kinase (CaMK) II have been observed to be activated by ROS in different experimental settings [192,193,194]. The p38 MAPK has been shown to activate myocyte enhancer factor MEF2 [195], which assembles with the co-activator PGC-1α in skeletal muscle [196]. Moreover, Handschin and colleagues collected evidence in favour of CaMK IV associating with cAMP binding response element (CREB) in order to induce PGC-1α transcription. Similarly, the catecholamine-induced adenylyl cyclase–PKA pathway leads to CREB activation and subsequent PGC-1α transcription [197]. Interestingly, viral transfection of C2C12 myoblasts with a dominant-negative MEF2C plasmid resulted in a declined PGC-1α expression, suggesting a positive feedback loop maintaining the oxidative adaptation [196]. The p38 MAPK can additionally regulate the transcription of PGC-1α through the activation of transcription factor 2 (ATF2) [198]. It can also directly promote PGC-1α activity through phosphorylation [199]. NF-κB appears to exert a negative regulation on the PGC-1α in the vasculature, as prenatal chronic inflammation and consequent NF-κB activation in rat foetus resulted in profound repression of PGC-1α target genes, leading to early onset hypertension [200]. NF-κB has been shown to directly associate with PGC-1α protein through the p65 subunit, following the activation by TNF-α [201]. This resulted in depressed PDK4 expression and elevated glycolytic flux in cardiomyocytes, which is characteristic of the aberrant hypertrophic heart. Thus, chronic inflammation due to either antioxidant inability in metabolic disease or overtraining and overwhelming the antioxidant capacity is detrimental to PGC-1α signalling. Notably, PGC-1α upregulation inhibited TNF-α-activated cytokines, including NF-κB and IL-6, in myotubes [202], highlighting the reciprocal relationship between inflammation, oxidative stress, and oxidative adaptation.

PGC-1α activation does not only promote metabolic proficiency and REDOX balance, but it also coordinates Ca^2+^ homeostasis. Inflammation-dependent PGC-1α inhibition resulted in a decline of PDK4 expression favouring enhanced glycolytic flux [201]. Downregulation of PGC-1α target proteins GLUT4, PPAR-α, CPT-1, and MCAD with PDK4, in particular, has been observed in calcineurin-induced cardiac hypertrophy [203]. Phosphatase calcineurin over-activation has been matched with the chronic intra-myofibrillar Ca^2+^ elevation in the hypertension-overloaded heart [204]. As a counter-image of the hypertensive phenotype, PGC-1α-overexpressing mice exhibited elevated mRNA levels of SR Ca^2+^ ATPase (SERCA2A), phospholamban (PLN), and sodium/calcium exchanger 1 (NCX1) [205]. PLN, a negative regulator of SR Ca^2+^ release, and NCX, a fast buffer dissipating cytosolic Ca^2+^ increments, contribute to stricter and improved Ca^2+^ handling [206]. Therefore, aiming to activate PGC-1α signalling pathways through regular exercise should promote metabolic flexibility as well as retardation of age-associated cardiomyopathy [205] and skeletal muscle sarcopenia [207,208].

Discriminating the one critical pathway responsible for inducing PGC-1α expression appears to be rather a “Gordian knot” task, with no clear-cut solution. Most likely, neither of the ROS, Ca^2+^, lactate or sympathetic stimulation is a dispensable pathway. Instead, they work in a unified network to generate a “perfect” transcriptional signal, which is essential considering a fine line between oxidative adaptation and oxidative overload. However, for the sake of the most accurate responses, it would be logical to employ more immediate strategies to induce nuclear translocation of the cytosolic PGC-1α pool [209]. Wright and colleagues, who subjected rats to 6 h of a swimming exercise, observed that the target transcription factors of PGC-1α, nuclear respiratory factor 1 (NRF1) and 2 (NRF2), were binding to their promoter sequences before any significant PGC-1α mRNA increment. Significant phosphorylation of p38 MAPK at the early stage of exercise suggested phosphorylation-induced PGC-1α translocation. Western blots of nuclear extracts from muscle biopsy, taken at the early stage of swimming, validated that claim [209]. Activation of cytosolic PGC-1α through phosphorylation has been also shown in humans during a 90-min cycling exercise at 65% of VO_2_ max [210]. Apart from p38 MAPK, AMPK has been identified as a potential acute pathway of PGC-1α activation, as acetyl-CoA carboxylase (ACC) phosphorylation (a marker of AMPK activity) was nearly five times the pre-exercise value.

## 10. The Higher the Intensity, the Larger the Dyshomeostasis, the Greater the Adaptation?

Consistent with the concept of metabolic inertia, AMPK is an excellent candidate to signal the energetic stress for oxidative adaptation. With a high AMP/ATP ratio in a vigorously contracting muscle fibre, AMP binds to regulatory γ subunit of AMPK, resulting in conformational changes in the heterotrimeric structure and phosphorylation of critical threonine and serine residues of the catalytic AMPK α subunit [211]. AMPK has been reported to indirectly activate NAD^+^-sensitive deacetylase SIRT1, which in turn allosterically activates PGC-1α, promoting mitochondrial biogenesis [212,213]. AMPK can also directly activate PGC-1α protein through phosphorylating the threonine-177 and serine-528 residues [214]. The degree of AMPK activation appears to be isoform-specific, varying with exercise intensity and duration. A 90-min cycle at 75% of VO_2_ max induced a nearly fourfold activation of α2 AMPK isoform, but no change in the activity of the α1 isoform was noted [215]. Intriguingly, neither α1 nor α2 AMPK isoform was activated following the same duration cycle at 40% of VO_2_ max. During a steady cycling exercise at 45% of VO_2_ max, lasting for more than 3 h until exhaustion, phosphorylation of threonine-177 progressively increased further into the trial. This was reflected by an increment in α2, but not α1, AMPK activity [216]. Interestingly, ACC-β phosphorylation was at the peak after 1 h but declined back to baseline toward the end of the prolonged trial. This suggests that AMPK-mediated phosphorylation does not promote adaptation to long-endurance exercise. However, in other exercise protocols including three 20-min long increments, both α1 and α2 activity was induced with a parallel increased trend of phosphorylation of ACC-β and neuronal nitric oxide synthase μ (nNOSμ) [217].

Interestingly, nNOSμ has been discovered to associate with dystrophin in a sub-sarcolemmal compartment, with DMD-associated dystrophin malfunction resulting in mislocalisation of nNOSμ and contractile dysfunction [218]. The aforementioned nNOSμ and another isoform nNOSβ have been recognised to promote exercise performance [219]. NOS has been reported to relieve the xanthine oxidoreductase (XOR) inhibition of cardiac excitation-contraction coupling [220]. A couple of studies presented strong evidence of nNOS promoting glycolytic flux in skeletal muscle [221,222]. Increased glycolysis, in turn, provides NADPH for NOX2 and NOX4 catalytic activity in actively contracting muscle. Similar to nNOS, NOX enzymes found in the sarcolemma, SR tubules, and T tubules have been implicated in promoting excitation-contraction coupling [223]. As ADP-induced state 3 of OXPHOS ROS production is surprisingly smaller than the resting level at state 4 [224], NOXs emerge as the main contributors of exercise-stimulated ROS generation, with NOX4 being a constitutively active isoform and NOX2 becoming more active during active muscle contractions [225]. Considering that NOX4 and NOX2, in particular, are preferably expressed in red oxidative fibres [226], one can dispute that nNOS bears a critical role in orchestrating the matching of metabolic demand (as AMPK regulates nNOS expression [227]) with ROS generation. This nNOS-provided (via NOX activation) ROS-to-metabolic efficiency matching is reflected by the fact that antioxidant treatment in ROS-overloaded DMD fibres returned force-generating capacity and metabolic enzyme activity to control levels [169]. This is consistent with the studies that connected NOX2 hyperactivity and/or chronic ROS elevation with insulin resistance and metabolic inflexibility in vascular endothelial cells [228] and C2C12 myotubes [229].

Another important function of NOS in exercise is the induction of vasodilation through activating the guanylyl cyclase–cyclic guanosine monophosphate-dependent protein kinase (PKG) pathway in smooth muscle cells of neighbouring vasculature [230]. The obvious significance of this mechanism relates to the need for oxygen delivery to actively oxidising, contracting muscles. However, high energy demand and VO_2_ can exceed the capacity of the vasculature to promote larger perfusion, leading to localised muscle hypoxia as PO_2_ falls even to 3–7 mmHg [30]. Arguably, tissue hypoxia is the ultimate stressor, which promotes oxidative adaptation, or if failed, a maladaptation and metabolic disease.

## 11. Hypoxia—Friend or Foe?

For several decades, exercise researchers were aware of muscle hypoxia in exercise. However, a potential explanation for how hypoxia might promote exercise adaptation came at the brink of the millennium with the discovery of the hypoxia-inducible factor 1 (HIF-1) [230]. This discovery provided a new perspective to previous studies that observed enhanced exercise adaptation when training in blood flow-restriction ischemia [231] and normobaric hypoxia [232]. Interestingly, HIF-1 is highly conserved among mammalian species [233] and is expressed in virtually all tissues, including skeletal muscle [234]. Two functional subunits are identified: constitutively active HIF-1β and HIF-1α that associates with the former upon hypoxic stimulus and coordinates transcriptional responses. HIF-1α is continuously targeted for degradation by α-ketoglutarate and O_2_ dependent hydroxylation with prolyl hydroxylases (PHD) [235]. HIF-1α has been associated with acute hypoxic signalling, as after a month of training one leg with knee extension exercise, only the untrained leg muscle biopsy revealed upregulated HIF-1α, immediately after two-legged exercise, suggesting that the preconditioned leg was less hypoxic than the untrained one [236]. In this study, Lundby and colleagues explained the attenuated HIF-1α response in the trained leg by elevated levels of HIF-2α, which has also been shown to promote adaptation to chronic hypoxia at high altitude [237]. Contrary to previous speculations, HIF-2α rather than HIF-1α promotes improved O_2_ delivery through inducing expression of vascular endothelial growth factor (VEGF) and erythropoietin (EPO) [238]. However, HIF-1α promotes glycolytic adaptation by upregulating the expression of phosphoglycerate kinase 1 (PGK1), LDHA, GLUT4, and PDK1 [239,240]. Moreover, Mason and colleagues observed that HIF-1α-deficient mice exhibited improved mitochondrial function and endurance performance, suggesting that acute exercise stress could be maladaptive unless it is a chronic training regime. Elite endurance athletes exhibit improved machinery of HIF-1α-negative regulation as they have significantly higher transcript levels of PHD2 (metabolism) and factor-inhibiting HIF (FIH) with SIRT6 (both inhibit transcriptional activity) than moderately active individuals. Additionally, a transcriptional target of HIF-1α, PDK1, was significantly less abundant in the athletes’ muscle biopsies. The same study reported similar transcriptional trends in moderately fit subjects following a 6-week training programme [241].

Although HIF-1α seems slightly redundant to endurance adaptation, animals lacking this protein were found to be extremely susceptible to oxidative stress and muscle damage caused by downhill running, which elicits particularly stressful eccentric loading [239]. HIF-1α has been speculated to directly inhibit the oxidative phenotype development by repressing PGC-1α [242] and c-myc [243,244], as well as inducing expression of mitophagic proteins BNIP3 and NIX [245], although the latter remains controversial [246]. Deceptively, it appears that HIF-1α has a beneficial antioxidant role by suppressing the mitochondrial electron transport chain [247,248]. On the other hand, with acute intermittent hypoxia exposure, HIF-1α promotes the expression of the major ROS contributor in exercising muscle, NOX2 [249]. NOX2 in turn promotes glucose uptake, beneficial for the glycolytic phenotype [250]. Moreover, intermittent hypoxia-induced NOX2 and XO ROS activated calpain proteases, which consequently targeted HIF-2α for degradation [251,252]. It has been suggested that XO-mediated uric acid synthesis and consequent NOX2 O_2_^−^ generation are essential for HIF-1α activation. Since AMP is the limiting substrate for the uric acid generation pathway, it marks a bridging point between the metabolic and hypoxic stresses which employ acute ROS signalling and HIF-1α activation.

The same kind of bridge between metabolic and hypoxic stressors could be articulated in the oxidative phenotype signalling, as PGC-1α has been shown to induce the transcription of HIF-2α (accentuated by NAD^+^ sensitive SIRT1), which in turn promotes oxidative fibre type development [253]. On top of this, NOX4 that is implicated in a baseline O_2_^−^ production [225], when selectively silenced with small-interfering RNA (siRNA) led to a significant decline in HIF-2α expression [254]. The potential significance of this comes in the paradigm of chronic hypoxia and endurance training regimes ameliorating HIF-2α oxidative phenotype. Thus, prolonged ROS generation if matched by the metabolic response promotes oxidative and antioxidant adaptation. By any means, HIF-1α is not redundant. It is particularly important in promoting glycolytic fibre adaptation to HIIT [186,255] and enhancing top-end power output in power/sprint athletes [256]. The emerging picture of hypoxic and metabolic adaptations presents a continuous competition between HIFs and their associated phenotypes of glycolytic (acute hypoxia) and oxidative (sustainable O_2_ delivery in steady state) traits. In an evolutionary context, HIF-2α tips the scale toward the oxidative entity, progressively ageing mitochondria, while HIF-1α balances the cells toward glycolytic, oxygen-independent proliferation and, on rare occasions, tumorigenesis.

## 12. The Balance between Regulated Ageing and Immortal Tumorigenesis—Losing the Oxidative Identity

Mitochondrial biogenesis has a particular significance in cancer [257], age-related neurodegenerative diseases [258], and the ageing process in general [259]. The aforementioned advocate of mitochondrial biogenesis, PGC-1α, regulates many transcriptional factors including TFAM, MEF2, ATF2, NRF1, and Nrf2. It is important to notice that NRF2 and nuclear factor erythroid-derived 2-like 2 (Nrf2) are encoded by two distinct genes, GABPA and NFE2L2, respectively [260]. NRFs and Nrf-2 promote mitochondrial biogenesis and antioxidant adaptation through the activation of the antioxidant response element (ARE)-dependent genes. NRFs appear obligatory for mitochondrial biogenesis, in contrast to upstream PGC-1α, which strikingly has been reported to be dispensable for exercise adaptation in muscle-specific knockout mice [261]. NRF1 has been identified as the crucial downstream effector, directly binding to ARE region in the promoter of TFAM [262]. Consistently, deletion of the PGC-1α binding sequence in NRF1 essentially abolishes the PGC-1α effect on mitochondrial biogenesis [42]. The NRFs contribute to the oxidative phenotype through distinct transcriptional targets. For instance, hepatocyte-specific NRF1 deletion leads to profound liver damage, regardless of the activation of several Nrf2-dependent genes [263]. NRF1 knockout mice had a particular deficiency in metallothioneins, which serve an antioxidant function by buffering free radicals and nucleophilic heavy metals [264]. NRF1 target genes are involved in mitochondrial biogenesis as well as extra-mitochondrial processes, including splicing, cell cycle, RNA metabolism DNA damage repair, protein translation initiation, and ubiquitin-mediated protein degradation [265]. Notably, amongst the sequenced NRF1 target genes were those implicated in age-related neurodegenerative Parkinson’s and Alzheimer’s diseases.

The importance of Nrf2 for orchestrating mitochondrial biogenesis has been highlighted with the identification of AREs in NRF1 promoter, responsive to Nrf2 binding [266]. Recent findings propose that Nrf2 also regulates the expression of the upstream PGC-1α, as Nrf2 muscle-specific deletion and C2C12 in vitro silencing both resulted in a significant downregulation of PGC-1α and consequently halted mitochondrial biogenesis [267]. The same year, Aquilano et al. [268] identified Nrf2-responsive ARE sequences in PGC-1α. Similar to other transcriptional factors, Nrf2 is localised in the cytosol, where constitutively bound Keap1 facilitates Nrf2 ubiquitination, targeting it for proteasomal degradation [269]. Tonic inhibition is relieved with ROS-mediated oxidation of Keap1 sulfhydryl group [270]. Consequently, Nrf2 translocates to the nucleus, where it promotes the transcription of genes involved in REDOX homeostasis, OXPHOS, FAO, and mitochondrial biogenesis [271]. Another negative regulator of Nrf2 is glycogen synthase kinase 3β (GSK3β), which phosphorylates and consequently marks Nrf2 for proteasomal degradation [272]. However, this negative regulation can be suppressed by PKB upon stimulation with growth factors [273]. The Akt/PKB pathway may also be inhibited with a tumour suppressor protein PTEN, which is deactivated by ROS-mediated oxidation [274]. Additionally, p38 MAPK tied in PGC-1α direct activation [199] and transcriptional upregulation through ATF2 [198] also relieves the GSK3β inhibitory activity on Nrf2 [275]. Therefore, in a grand scheme, PGC-1α and Nrf2 establish a positive feedback loop promoting mitochondrial biogenesis and OXPHOS in response to exercise-induced ROS (Figure 4).

Interestingly, tumour suppressor p53 has been shown to activate the PGC-1α/Nrf2 pathway to induce SOD2 expression in response to REDOX dyshomeostasis or calorie restriction [268], highlighting the paradigm of anti-tumorigenic development toward the oxidative phenotype. PGC-1α and Nrf2 can be simultaneously activated in the Erk1/2 pathway, which includes first liver kinase B1 (LKB1) and AMPK subsequent phosphorylations, inducing neuroprotective mitochondrial biogenesis in the hypoxic hypothalamus [277]. An observation of HIF-1α-driven tumorigenesis and aerobic glycolysis in LKB1-deficient cells [278] adds to the significance of the above Erk1/2 pathway as pro-oxidative as well as anti-tumorigenic. LKB1 may act as a “memory” molecule aiming to maintain the established pro-oxidant feedback loops, as LKB1 and PGC-1α mRNA levels have been shown to consistently rise throughout a 53-day training regime [279]. Notably, acute lactate administration has been shown to activate hypertrophy-associated Akt/mTORC1 and Erk1/2 pathways in glycolytic fibres in mice [280]. However, the glycolytic fibres represent the exception to the homeostatic rule, as a vast majority of tissues aim to remain oxidative and energy-efficient. Most cells do not need to generate incredibly high yet unsustainable metabolic rates and they are especially not destined to proliferate in response to acute hypoxia and HIF-1α signalling. Cerda-Kohler et al. [280] also reported a lactate-induced elevation in AMPK and ACC phosphorylation in more oxidatively dominant muscles, implying an adaptive role of aerobic exercise in oxidative tissues like the brain. Steady-state submaximal exercise would be expected to elicit the AMPK/PGC-1α and Nrf2 signalling pathway, consistent with observations made by Hota et al. [277]. This hypothesis is supported by studies reporting increased muscle mitochondrial density due to upregulated expression of Nrf2 in animals subjected to an endurance training regime [281,282]. Moreover, exercise-induced activation of Nrf2 has been shown to attenuate pharmacologically induced hemi-Parkinsonism in mice by inducing mitochondrial biogenesis in nigrostriatal neurons [283].

Notably, Erk1/2 has been shown to phosphorylate HIF-2α, masking the nuclear export signal (NES) sequence that is recognised by nuclear exporter chromosomal region maintenance 1 (CRM1), therefore promoting HIF2α nuclear accumulation and transcriptional activity [284]. EPO upregulated by sustained hypoxia in a HIF-2α-dependent manner has been speculated to activate PGC-1α and Nrf2 pathways in multiple settings. EPO-induced activation of the pro-oxidative pathways ameliorated secondary brain injury following trauma in mice [285]. It significantly slowed down cognitive deficits in ageing rats [286] as well as suppressed myocardial apoptosis and inflammation in a sepsis-induced heart injury [287]. Finally, in skeletal muscle, EPO induced fibre-switching to slow-twitch and upregulated the OXPHOS, resulting in a doubled oxygen consumption [288]. Conclusively, exercise-associated chronic yet physiological oxidative stress should promote oxidant–antioxidant balance in body tissues, resulting in declined inflammation, mitochondrial density loss, and apoptosis, which are the hallmarks of ageing cells.

For many years, it has been thought that the pro-oxidative development, although distancing most cells from tumorigenesis, leads to progressive oxidative damage apparent in ageing [289]. This “free radical theory of ageing” is not wrong but does not present the complete picture. Cells such as muscles or neurons that do not recycle, but have a prolonged lifespan, do not remain constant in their mitochondrial profile. A continuous cycle of organelle turnover is dependent on FOXO transcription factors, coordinating proteolytic pathways, ubiquitin–proteasome with autophagy–lysosomal, and mTORC1 cascade associated with protein translation and autophagy inhibition [290]. The current perspective on ageing suggests that senescence-related increment in oxidative damage is caused by impaired autophagy of stressed mitochondria [259]. As recent literature proposes, AMPK-induced mitophagy is an essential component for muscle adaptation to endurance exercise [291,292,293]. One can easily conclude a suitable application for regular endurance exercise to improve age-related dysregulation of mitochondrial recycling. Stress-activated AMPK promotes mitophagy through activating critical autophagosome constituent ULK1. Additionally, phosphorylated AMPK activates tumour-suppressing TSC-2 that inhibits mTOR, which not only promotes translation and proliferation but also disrupts autophagosome assembly [293]. Therefore, regular endurance exercise promoting mitochondrial biogenesis through PGC-1α/NRF2 pathways as well as AMPK-mediated mitophagy should allow one to strike a golden medium of “regulated ageing”. This assumption is consistent with the observation of extended lifespan in *Caenorhabditis elegans* nematode due to pharmacological stimulation of SKN-1 (Nrf2 ortholog), which promotes both mitochondrial biogenesis and mitophagy [294]. Importantly, just as PGC-1α/NRF2 pathways promote upregulation of the ETC constituents, HIF-1α has an opposite effect [248], which proved to be cytoprotective, as lidocaine-induced ETC disruption and caspase activation was inhibited in HIF-1α-overexpressing cells, rendering the cells resistant to apoptosis [247]. This reflects the association of HIF-1α and synonymous intermittent hypoxia, inflammation, and lactate accumulation with the immortal phenotype of cancer.

As high-impact resistance and high-intensity modalities of exercise promote proliferative mTORC [295] and glycolytic HIF-1α signalling [185,255], they should rather be avoided or carefully implemented from the perspective of a senile or metabolically compromised person. Moreover, HIIT, in contrast to endurance regimes, did not relieve cardiac fibrosis and promoted additional ventricular hypertrophy in hypertensive rats [185]. Again, HIIT is particularly advantageous for well-developed athletes and should be preceded by a period of moderate preparation training to allow the system to super-compensate the oxidative damage with a more efficient metabolism and antioxidant system [164]. A notable characteristic of the contrasting metabolically inflexible phenotype is lactate accumulation very early into even mild exercise [137]. Notably, immortal cancer cells adapt an extremely glycolytic phenotype, which is synonymous with lactate accumulation. However, similar to contracting muscle fibres, cancer cells often shuttle lactate in their tumour “microenvironment” in-between surprisingly oxidative tumour cells and glycolytic cancer-associated fibroblasts (CAF), which take over as the main lactate producer in a “reverse Warburg effect” under nutrient scarcity [296]. During the early stages of tumorigenesis, neoplastic cells promote CAF’s glycolytic ability through ROS-mediated mitophagy and the abolition of OXPHOS capacity [297]. This leads to the emergence of an updated model of tumorigenesis, suggesting that oncogenic mutations allow neoplastic cells to harness mitochondrial metabolism for ROS production to induce CAFs, which in turn provide lactate for proliferation [298]. However, a question remains as to why the tumour-potent cells are not damaged by this extensive ROS generation? A confounding yet obvious answer is an improved antioxidant capacity. The same lactate that promotes adaptation in exercise induces a moderate burst of ROS, leading to pro-antioxidant Nrf2 activation as well as stimulation of proliferative Akt and the unfolded protein response, which allows for an elevated protein synthesis capacity [299]. Therefore, the oncogenic disease does not necessarily mean a phenotypic shift from oxidative to glycolytic as it has been believed for decades after the discovery of the Warburg effect. Just as in all cells of the body, ROS-sensitive pathways coordinate a functional balance between glycolytic and oxidative. The oncogenic mutations cause alterations in these regulatory loops, leading to the establishment of a novel balance that is functional from the perspective of the archaic “protoeukaryote”, which learns through mutation how to harness the power of OXPHOS. Although mitochondria provide an astounding capacity for the endosymbiotic eukaryote to oxidise carbon sources for vast amounts of energy, it has a price for its service. The price is obvious and inevitable, it is called ageing. This evolutionary co-dependence is rooted in the genetic code, as Nrf2 has been found to recognise ARE sequence in the HIF-1α promoter region [300], hence pinpointing the centre of the oxidative versus glycolytic fulcrum. In the case of metabolic disease, Nrf2 sits too near to the fulcrum and becomes outweighed by HIF-1α. With adequately balanced oxidative stress, Nrf2 can be moved back to its side of the balance swing to attain better leverage for the oxidative phenotype. However, in the case of carcinogenic mutation, Nrf2 (or mutated components of the pro-oxidative loop) sits on the completely wrong side of the swing, still maintaining expression of the oxidative phenotype genes, but this time in favour of the opposing team—glycolytic proliferation (Figure 5).

## 13. Summary

In summary, millions of years of eukaryotic evolution armed the anaerobic ancestor with a powerful energy hub, the mitochondrion [301]. However, this oxidative symbiont needs constant reassurance by the system. If not stimulated, it forgets how to do the job it has been doing for millennia. Under-using the oxidative entity leads to the dominance of the glycolytic phenotype, which cannot handle the oxidative and macronutrient stress, leading to a broad palette of associated disorders including type 2 diabetes, obesity, hypertension, and heart failure. As the mitochondria lose their oxidative identity, they accumulate without the ability to rejuvenate. Consequently, the golden medium between mitophagy and biogenesis becomes unattainable, thereby progressing the organism into senescence. Endurance exercise emerges as a universal magic pill to settle the quarrels between the oxidative and anaerobic symbiont. Carefully calibrated endurance exercise regimes will stimulate the oxidative entities through multiple cascades (including ROS) and teach the system how to coordinate the nutrient fluxes into the oxidative hub, and progressively will return the metabolic flexibility and free the tissues of pathological hypoxia and inflammation. Unfortunately, in the last century, the oxidative–glycolytic metabolic relationship has deviated into mutated ways, which manifest in proliferative and immortal cancer. Perhaps glycolysis is the true archaic way, with the endosymbiosis and oxidative evolution steering the “protoeukaryote” away from it. It is possible that we are losing our oxidative way, with every new generation coming to this oxygen-rich world disadvantaged. This appears to be true, with a multitude of studies showing that a lack of physical activity in both mother and father leads to maladaptive epigenetic modifications in pro-oxidative genes and consequently compromised phenotype in the offspring [302,303,304,305,306].

## 14. Conclusions

The ability of humans to efficiently harness the chemical energy from macronutrients through oxidative metabolism puts our species on an evolutionary pedestal. A well-orchestrated carbon flux into mitochondria in a syncytium with cytosolic glycolysis can also be a powerful tool for endurance exercise performance of various modalities and durations. However, the downfall of the oxidative character of our cells is the need for a consistent updating of the mitochondrial contents by exposing the cells to an adaptive dose of hormesis or oxidative “eustress” [307]. Skeletal muscles, as arguably the most metabolically powerful system, need to be stimulated. This can be achieved by regular endurance exercise [308], and intensities should be tailored accordingly to the individual’s oxidative phenotype. Endurance exercise emerges as a universal medicine to treat and prevent the most prevalent diseases of the modern Western world. These include the umbrella of metabolic syndrome, cancer, and neurodegenerative diseases, all of which are characterised by chronic oxidative stress. As epigenetic maladaptations render the following generations more prone to disease, the global healthcare system may become drastically overloaded. Therefore, raising awareness about the extensive health benefits of endurance exercise should become a priority in developed countries.

## Figures and Tables

**Figure 1 antioxidants-10-00609-f001:**
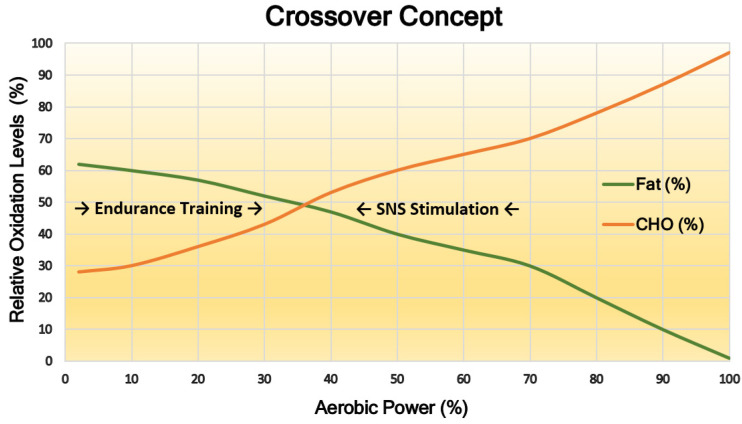
The crossover concept. Adapted from Brooks and Mercier [26].

**Figure 2 antioxidants-10-00609-f002:**
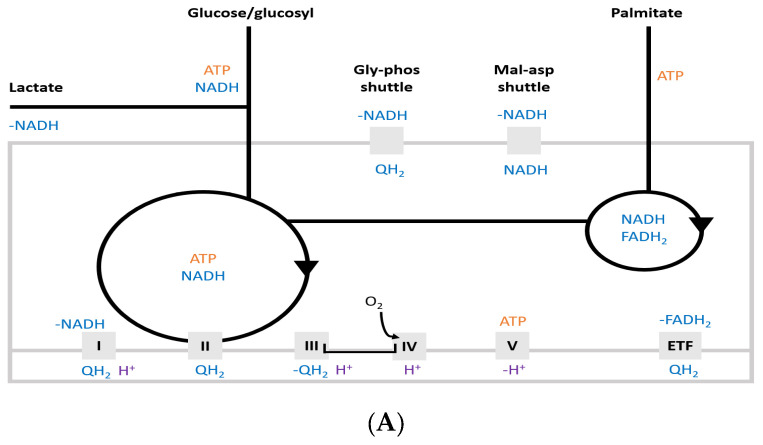
(**A**) Graphic representation of intermediary energy metabolism including carbon fluxes, energy substrate oxidation (NADH generation) sites, and electron transport chain. (**B**) Metabolic flux trade-off between substrate efficiency (ATP cmol^−1^; *y*-axis), and catalytic capacity (mmol ATP [g of protein enzyme]^−1^ h^−1^; *x*-axis). Adapted from Nilsson et al. [27].

**Figure 3 antioxidants-10-00609-f003:**
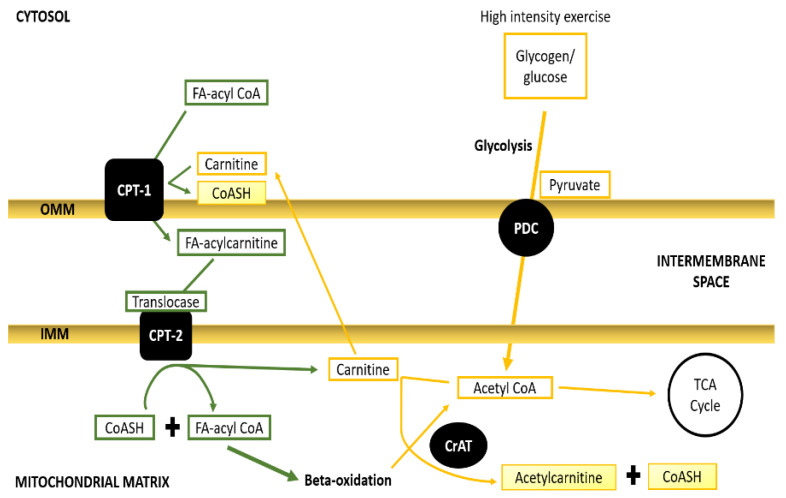
Fatty acid acyl-CoA–carnitine shuttle. Adapted from Jeppesen and Kiens [82]. Carnitine palmitoyl transferase (CPT)-1 mediates fatty acid (FA) acyl-CoA interconversion with free carnitine to FA-acylcarnitine, promoting transport into intermembrane space. The translocase enzyme facilitates FA-acylcarnitine entry into the mitochondrial matrix. CPT-2 regenerates free carnitine and FA acyl-CoA, which now can be oxidised. Carnitine acetyltransferase (CrAT) is a matrix enzyme with two polarities. The reverse reaction (presented above) breaks down acylcarnitine to feed acetyl-CoA into tricarboxylic acid (TCA)/Krebs cycle (particularly during metabolic inertia). When glycolytic flux through pyruvate dehydrogenase complex (PDC) is matched to a new steady state, CrAT favours the forward reaction consuming free carnitine for acylcarnitine regeneration to mop the excess acetyl groups that would otherwise inhibit PDC activity. Hence, progressively increasing PDC flux might lead to inhibition of FA-carnitine shuttle via CrAT forward reaction.

**Figure 4 antioxidants-10-00609-f004:**
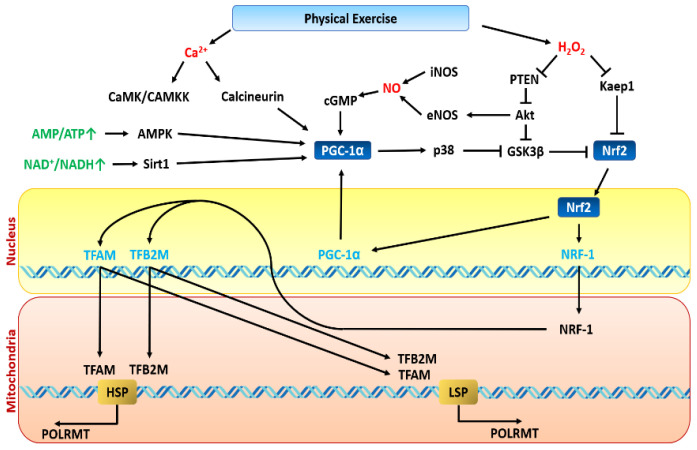
Oxidative transcriptional regulatory loop. Adapted from Gureev et al. [276]. Exercise promotes reactive oxygen species (ROS) generation, which leads to the relief of tonic inhibition of Keap1 on Nrf2. Nrf2 is also activated in a PGC-1α dependent mechanism. Nrf2 promotes NRF1 (enables the transcription of mitochondrial transcription factors, mitochondrial transcription factor A (TFAM) and transcription factor B2, mitochondrial (TFB2M)) and PGC-1α, establishing a positive feedback loop of pro-oxidative signalling in response to exercise-induced increments in ROS, NO, Ca^2+^, adenosine monophosphate (AMP), and oxidized nicotinamide adenine dinucleotide NAD^+^.

**Figure 5 antioxidants-10-00609-f005:**
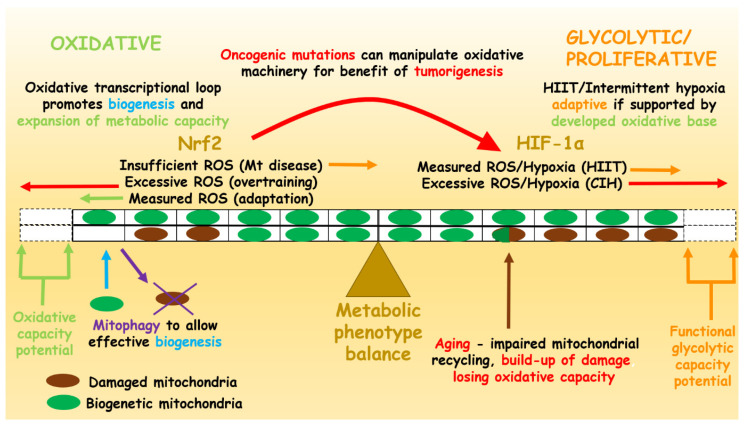
Model of metabolic phenotype balance. Light green arrows indicate pro-oxidative adaptation, yellow—pro-glycolytic/proliferative adaptation, and red—maladaptation due to over-load of endogenous antioxidant system or malignant ROS signalling in cancer. Mt: metabolic; HIIT: high-intensity interval training; CIH: chronic intermittent hypoxia.

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
