# Peer review of "Re-Evaluating the Oxidative Phenotype: Can Endurance Exercise Save the Western World?"

_antioxidants, 2021, doi:10.3390/antiox10040609_

Round 1

Reviewer 1 Report

This article is very interesting, comprehensive and well explaining the tricky explanation of oxydative stress balance during heavy endurance exercise. 

It would be even more interesting if the authors could speculate on  the effect of endurance training in high risk subjects, but I suppose that literature is very poor on this speciphic field. 

Author Response

Reviewer 1

This article is very interesting, comprehensive and well explaining the tricky explanation of oxydative stress balance during heavy endurance exercise. 

It would be even more interesting if the authors could speculate on the effect of endurance training in high risk subjects, but I suppose that literature is very poor on this speciphic field. 

Response: We thank the reviewer for the positive comments. We have provided commentary on the preferred mode of training in disease.

Reviewer 2 Report

Filip Kolodziej and Ken O’Halloran present an extensive review on endurance exercise and mitochondrial/metabolic adaptations that occur with aerobic training that may prevent several disease states. The review is thorough and does an excellent job of taking the reader through the literature. It was a pleasure to read and I only have some minor suggestions that could improve the quality of the paper. 

  • In the abstract, first sentence, consider just starting with the word "Mitochondria...". The way it is worded (the mitochondria) sounds a bit awkward. 
  • In the abstract, line 14, consider changing the word "metabolism" with "capacity" to be consistent with the rest of the abstract. 
  • In the abstract, line 22, consider changing the word "develops" to "improves".
  • On line 99, the sentence "Second, ATP needs to be provided for the myosin head....". I understand what the author means here; however, it is more accurate to say it is needed to break the connection between myosin and actin (preventing rigor). 
  • Line 119, lactate is also thought of being a substrate that dictates glycolytic versus aerobic pathways. Consider adding a sentence on this from the recent work of George Brooks (lactate shuttle theory). High fasting plasma lactate is also thought to be a marker of metabolic disease as it has been show to be higher in individuals with obesity/type 2 diabetes (see recent work from Houmard, Dohm, Pories). This might be useful to add since it fits with some of the themes on lactate and metabolic flexibility and, as noted, lactate is historically thought of as just a waste product. 
  • Figure 2 is extremely confusing and took me several reads to be able to interpret the legend. Please consider simplifying. For example, why are there 6 squares for the ETC component? Should it not be 5 for the complexes? 
  • Line 223, is "Fat Min" defined somewhere? 
  • Lines 883 and 885 seem to having spacings issues between sentences. 

Author Response

Reviewer 2

Filip Kolodziej and Ken O’Halloran present an extensive review on endurance exercise and mitochondrial/metabolic adaptations that occur with aerobic training that may prevent several disease states. The review is thorough and does an excellent job of taking the reader through the literature. It was a pleasure to read and I only have some minor suggestions that could improve the quality of the paper. 

  • In the abstract, first sentence, consider just starting with the word "Mitochondria...". The way it is worded (the mitochondria) sounds a bit awkward. 
  • In the abstract, line 14, consider changing the word "metabolism" with "capacity" to be consistent with the rest of the abstract. 
  • In the abstract, line 22, consider changing the word "develops" to "improves".
  • On line 99, the sentence "Second, ATP needs to be provided for the myosin head....". I understand what the author means here; however, it is more accurate to say it is needed to break the connection between myosin and actin (preventing rigor). 
  • Line 119, lactate is also thought of being a substrate that dictates glycolytic versus aerobic pathways. Consider adding a sentence on this from the recent work of George Brooks (lactate shuttle theory). High fasting plasma lactate is also thought to be a marker of metabolic disease as it has been show to be higher in individuals with obesity/type 2 diabetes (see recent work from Houmard, Dohm, Pories). This might be useful to add since it fits with some of the themes on lactate and metabolic flexibility and, as noted, lactate is historically thought of as just a waste product. 
  • Figure 2 is extremely confusing and took me several reads to be able to interpret the legend. Please consider simplifying. For example, why are there 6 squares for the ETC component? Should it not be 5 for the complexes? 
  • Line 223, is "Fat Min" defined somewhere? 
  • Lines 883 and 885 seem to having spacings issues between sentences. 

Response:  We thank the reviewer for the positive comments and constructive critique of our manuscript. We used your recommendations to improve the manuscript’s quality and clarity (bullet point 1-4). Concerning section 8. ”Lactate sets the stage for PGC-1α signalling – the oxidative booster”, we think we made it clear enough that lactate should no longer be conceived as a metabolic waste product. Fitting with your recommendation of using George Brooks’ pioneering work, we reference his recent study in collaboration with Inigo San Millan. They carried out a cycling GXT in three different populations: Type 2 diabetics, recreationally active/amateur athletes and professional cyclists racing at the highest level in the UCI World Tour. We emphasise the observation that metabolically diseased individuals exhibit a rapid appearance of blood lactate at small work rates, which mirrored the rate of CHO oxidation, indicating that this population does not recycle lactate but must engage the fast glycolytic flux based on a high metabolic demand relative to their level of fitness. On the opposite end of the spectrum, the professional cyclists’ measurements work provide a neat proof of concept, as their lactate appearance is minimal for a much longer duration into the incremental protocol, indicative of active recycling. As soon as their blood lactate rises past the inflection point, the rate of increase begins to mirror the CHO oxidation, thus implying that lactate mitochondrial flux is a significant contributor to the production of aerobic power in endurance trained individuals, and that past critical work rate (which occurs very early in diabetics, but late in cyclists), glycolytic flux becomes the dominant supplier of chemical energy to mitochondria.

Regarding the Fat Min, it is defined by Achten and Jeukendrup who standardised the Fat max protocol and popularised the term. (Reference 9. Achten, J.; Jeukendrup, A. Maximal Fat Oxidation During Exercise in Trained Men. International Journal of Sports Medicine 2003, 24, 603-608.)

Considering the lack of clarity in figure 2, we have redrawn another figure from the referenced paper Nilsson et al. The additional figure is a schematic representation labelling the carbon fluxes, the sites of energy substrate oxidation / NADH generation and the electron transport chain components. This should help with the interpretation of the miniature schemes accompanying the Metabolic Modes model (now Figure 2 B) and understanding of the substrate efficiency – catalytic (ATP) capacity.

Figure 2. A) Graphic representation of intermediary energy metabolism including carbon fluxes, NADH generation sites and electron transport chain. B) Metabolic flux trade-off between substrate efficiency (ATP cmol-1; y-axis), and catalytic capacity (mmol ATP [g of protein enzyme]-1 h-1; x-axis). Adapted from Nilsson et al. [27].

Four optimal pathways (full orange circles) include the most efficient beta-oxidation through glycerol-3-phosphate shuttle and complex I bypass to fermentative glycolysis (does not enter TCA cycle, no ETC; left in grey). The sub-optimal pathways (open orange circles) are malate-aspartate shuttle and hydrogen uncoupling (does not contribute to ATP synthesis). Glycolytic and lactate fluxes feed carbon and electrons into the TCA cycle (large) and NADH shuttles with fatty-acid supply the alternative NADH pool (small). These energy flows are marked in blue if engaged in one of the optimal modes. Squares below the cycles represent the mitochondrial complexes 1-5 and electron-transferring flavoprotein dehydrogenase (ETF) as per figure 2 A) scheme. Complex I bypass shifts the curve right (dark blue), indicating higher substrate efficiency at high ATP synthesis/catalytic rates, relative to the alternative model (light blue).

Reviewer 3 Report

The authors are applauded for undertaking this interesting review evolutionary considerations of metabolism as related to aging and oxidative stress. Overall, this is an outstanding review, insightful and with impact. I have a handful of editorial suggestions to consider to improve reader understanding. I think there are arguments against the idea that the oxidative phenotype is changing due to physical inactivity in that the species is perpetuated despite the recent “blip” in physical activity and body composition. Moreover, my distant understanding about atmospheric changes and evolutionary considerations of conserved metabolism may not intertwine with the thrust of this review. That said, the manuscript is exceedingly compelling and should be published as a well thought out and well-articulated discussion starter. The manuscript is a bit dense, but by design (and perhaps necessity). I’ve made some recommendations to help the novice reader along. These comments are intended to be additive.

Abstract line 20: “this uncontrolled oxidative stress” I agree with the point you’re making here, but since this is a paradigm shift for many, consider editing to bridge common misconceptions about the topic, which may get confused with the idea that chronic high dose oxidative stress (e.g., HIV+ and cocaine use) can accelerate disease processes.

Line 190 and 689 – do you mean associate rather than correlate? If correlate, please report r or r2, etc., otherwise change to associate.

Consider bolstering the point about exercise metabolism and ROS production by highlighting important observations that the appearance of blood born oxidative damage markers following various exercise bouts is often disconnected from the total metabolic flux, but rather can be intensity dependent as found in:
Generation of reactive oxygen species after exhaustive aerobic and isometric exercise.

Alessio HM, Hagerman AE, Fulkerson BK, Ambrose J, Rice RE, Wiley RL.Med Sci Sports Exerc. 2000 Sep;32(9):1576-81. doi: 10.1097/00005768-200009000-00008.PMID: 10994907

The effects of acute exercise on neutrophils and plasma oxidative stress.

Quindry JC, Stone WL, King J, Broeder CE.Med Sci Sports Exerc. 2003 Jul;35(7):1139-45. doi: 10.1249/01.MSS.0000074568.82597.0B.PMID: 12840634

General editorial comments: The paper is very dense, although not without good effect. However, my impression is that it’s going to be well received by readers that are already well versed on the topic, while it may be too dense for less informed readers. Consider breaking long paragraphs with simple declarative sentences to provide the reader with a roadmap for understanding the introduction of various concepts.

Instances of passive language were noted, minor and fixable.

Line 568 – “proved” many still oppose this usage in hypothesis driven research venues – consider changing

Line 688 – “reduction of ROS” – indicate for the reader that you mean molecular reduction vs a mean decrease response. In context I suspect you mean the former, but some novice readers may not follow.

Consider inserting and referencing related key concepts from the following review:

The anaerobic threshold: 50+ years of controversy.

Poole DC, Rossiter HB, Brooks GA, Gladden LB.J Physiol. 2021 Feb;599(3):737-767. doi: 10.1113/JP279963. Epub 2020 Nov 19.PMID: 33112439

Author Response

Reviewer 3

The authors are applauded for undertaking this interesting review evolutionary considerations of metabolism as related to aging and oxidative stress. Overall, this is an outstanding review, insightful and with impact. I have a handful of editorial suggestions to consider to improve reader understanding. I think there are arguments against the idea that the oxidative phenotype is changing due to physical inactivity in that the species is perpetuated despite the recent “blip” in physical activity and body composition. Moreover, my distant understanding about atmospheric changes and evolutionary considerations of conserved metabolism may not intertwine with the thrust of this review. That said, the manuscript is exceedingly compelling and should be published as a well thought out and well-articulated discussion starter. The manuscript is a bit dense, but by design (and perhaps necessity). I’ve made some recommendations to help the novice reader along. These comments are intended to be additive.

Abstract line 20: “this uncontrolled oxidative stress” I agree with the point you’re making here, but since this is a paradigm shift for many, consider editing to bridge common misconceptions about the topic, which may get confused with the idea that chronic high dose oxidative stress (e.g., HIV+ and cocaine use) can accelerate disease processes.

 Line 190 and 689 – do you mean associate rather than correlate? If correlate, please report r or r2, etc., otherwise change to associate.

 Consider bolstering the point about exercise metabolism and ROS production by highlighting important observations that the appearance of blood born oxidative damage markers following various exercise bouts is often disconnected from the total metabolic flux, but rather can be intensity dependent as found in:
Generation of reactive oxygen species after exhaustive aerobic and isometric exercise.

Alessio HM, Hagerman AE, Fulkerson BK, Ambrose J, Rice RE, Wiley RL.Med Sci Sports Exerc. 2000 Sep;32(9):1576-81. doi: 10.1097/00005768-200009000-00008.PMID: 10994907

The effects of acute exercise on neutrophils and plasma oxidative stress.

Quindry JC, Stone WL, King J, Broeder CE.Med Sci Sports Exerc. 2003 Jul;35(7):1139-45. doi: 10.1249/01.MSS.0000074568.82597.0B.PMID: 12840634

Response: We are grateful for the positive comments and recommendations. We agree that all are valuable in helping to improve the clarity of this review, augmenting the impact.

Regarding the novel understanding of oxidative stress, we decided to improve readers’ understanding by use of the term “oxidative eustress” with citation to a 2021 review of a pioneer in this field, Helmut Sies:

Chronic overload of these reactive oxygen species (ROS) damages cell components like DNA, cell membranes, proteins etc. Counterintuitively, transiently generated ROS during exercise contribute to adaptive REDOX signalling through the process of cellular hormesis or ‘oxidative eustress’ defined by Helmut Sies [303]. However, the unaccustomed, chronic oxidative stress is central to the leading causes of mortality in the 21st century: metabolic syndrome and the associated cardiovascular comorbidities.

Regarding the correlation, we have added the r value at line 200:

Goodpaster et al. [38] reported a significant correlation (r = -0.57) between the histochemically quantified intramuscular lipid content and insulin sensitivity determined by the hyperinsulinemic-euglycemic clamp method.

And we changed the wording at the line 700 as the studies we refer to did not calculate statistical correlations between the measures of the expression of PGC-1α, SOD2, GPX1 with ROS emission and mitochondrial energy coupling:

The antioxidant adaptation is mediated by the aforementioned PGC-1α, whose overexpression resulted in increased antioxidant SOD2 and GPX1 mRNA, and reduction in ROS accumulation, paralleled by the diminished electron leak and elevated intermembrane space potential [178-180].

Although we were hesitant to extend the manuscript, we agree that your point on the mismatch between the inflammatory response and oxidative stress markers would enhance the readers’ understanding as well as enforce our proposal for careful/progressive administration/avoidance of HIIT in severely compromised phenotypes (section 12). We highlighted the critical role of exercise intensity for eliciting a measurable oxidative stress by referring to several recent cross-sectional studies at lines 705-722 (dating 2017-2020):

Although, the inflammatory response and oxidative stress are often considered to go hand in hand, several studies suggested that the appearance of inflammatory and oxidative stress markers are independent of each other, as vigorous walking increased IL-6 and TNF- α, but not markers of oxidative stress immediately post-exercise [304]. Similarly, inflammatory cytokines but not markers of DNA oxidation were elevated in runners, minutes after completing a marathon [305]. On the contrary, DNA oxidation markers were significantly increased immediately after and for another 90 minutes in half-marathon competitors [306]. Moreover, Jamurtas et al. [307] reported that only 4x30 seconds all-out cycling protocol is sufficient to elevate protein carbonylation in comparison to baseline. Notably, the same study exemplified the supremacy of short bouts of high-intensity relative to continuous 30 minutes of cycling at 70% of VO2 max, in terms of inducing the immune response and oxidative stress. However, high-intensity interval training (HIIT) should be avoided or carefully supplemented to the moderate endurance exercise training regime of diseased individuals, in this way ameliorating the compromised phenotype while minimizing the possibility of diminishing returns due to acute oxidative stress on top of chronic stress. This notion was supported by studies where in contrast to beneficial moderate endurance training, HIIT exacerbated cardiovascular disease in hypertensive rats [248, 290].